# Designing a Novel Approach Using a Greedy and Information-Theoretic Clustering-Based Algorithm for Anonymizing Microdata Sets

**DOI:** 10.3390/e25121613

**Published:** 2023-12-01

**Authors:** Reza Ahmadi Khatir, Habib Izadkhah, Jafar Razmara

**Affiliations:** Department of Computer Science, University of Tabriz, Tabriz 5166616471, Iran; r.ahmadikhatir@tabrizu.ac.ir (R.A.K.); razmara@tabrizu.ac.ir (J.R.)

**Keywords:** information theory, entropy, data anonymization, clustering, privacy-preserving, individuals’ privacy

## Abstract

Data anonymization is a technique that safeguards individuals’ privacy by modifying attribute values in published data. However, increased modifications enhance privacy but diminish the utility of published data, necessitating a balance between privacy and utility levels. *K*-Anonymity is a crucial anonymization technique that generates *k*-anonymous clusters, where the probability of disclosing a record is 1/k. However, *k*-anonymity fails to protect against attribute disclosure when the diversity of sensitive values within the anonymous cluster is insufficient. Several techniques have been proposed to address this issue, among which *t*-closeness is considered one of the most robust privacy techniques. In this paper, we propose a novel approach employing a greedy and information-theoretic clustering-based algorithm to achieve strict privacy protection. The proposed anonymization algorithm commences by clustering the data based on both the similarity of quasi-identifier values and the diversity of sensitive attribute values. In the subsequent adjustment phase, the algorithm splits and merges the clusters to ensure that they each possess at least *k* members and adhere to the *t*-closeness requirements. Finally, the algorithm replaces the quasi-identifier values of the records in each cluster with the values of the cluster center to attain *k*-anonymity and *t*-closeness. Experimental results on three microdata sets from Facebook, Twitter, and Google+ demonstrate the proposed algorithm’s ability to preserve the utility of released data by minimizing the modifications of attribute values while satisfying the *k*-anonymity and *t*-closeness constraints.

## 1. Introduction

Data privacy protection is a critically important area of research, especially as data collection from individuals increases and proliferates rapidly. Data often contains sensitive information that could pose privacy risks if released without proper processing. To ensure the privacy of individuals, many countries have enacted privacy protection laws, such as the General Data Protection Regulation (GDPR) in the European Union, the Personal Data Protection Act (PDPA) in Singapore, and the Personal Information Protection Law (PIPL) in China. These laws aim to regulate data handling practices and safeguard individuals’ personal information. As data collection and usage continue to grow exponentially, effective privacy protection techniques remain vital to uphold individuals’ rights while enabling data-driven innovation.

Preserving data privacy presents several challenges, the most significant being the prevention of information leakage during data release. Data can contain different types of information; therefore, it is crucial to identify confidential data and assess the potential consequences of a data breach. Identifying and classifying sensitive data are key steps in developing an effective data privacy protection strategy. Once the sensitive data has been identified, organizations can determine the appropriate data privacy protection measures to implement. To protect individuals’ privacy, it may be necessary to anonymize the data while considering the type of data, the intended use, and the potential privacy risks.

Data anonymization is a practical and widely adopted approach to protecting individual privacy when releasing data. Anonymization techniques aim to reduce the risk of unauthorized disclosure by converting personal information into anonymous data. This balanced approach to safeguarding privacy mitigates the original details in the dataset to some extent [1]. Anonymizing data involves modifying quasi-identifier values to reduce specificity, enhancing privacy protection while preserving data utility [2].

Data containing confidential information about individuals is prone to privacy breaches, so it is essential to protect sensitive information from privacy threats [3]. One way to protect personal privacy is to erase direct identifier attributes, such as names and addresses, before releasing the original data to the public. This process, known as de-identification, is insufficient to protect privacy, as a study by Sweeney [4] showed that 87% of Americans can still be uniquely identified based on just three quasi-identifiers: gender, age, and postal code. To prevent privacy threats, researchers have proposed several privacy models, including *k*-anonymity (KA) [5], *l*-diversity (LD) [6], and *t*-closeness (TC) [7]. In the following, we briefly overview these models and their anonymization operations.

Anonymization techniques have become increasingly important for ensuring privacy in data publishing. Among these techniques, KA is widely used by researchers [5]. KA aims to protect privacy by modifying quasi-identifiers to ensure that each record is indistinguishable from at least *k*-1 other records. It is important that individual records remain faithful to the original data while meeting privacy requirements after anonymization [8]. Generalization, suppression, and perturbation are the main anonymization operations used to achieve k-anonymization algorithms:Generalization replaces some quasi-identifier values with parent values in the equivalent class, while specialization is the opposite operation. Generalization preserves data privacy but reduces data accuracy [3]. Excessive generalization results in significant information loss (IL) and reduced data quality [9].Suppression replaces specific quasi-identifier values with asterisks (*), while the disclosure is the opposite. Suppression causes more IL than generalization. For example, suppose we have a dataset of people’s ages. We could generalize this dataset by replacing all ages with a range of ages, such as 20–30, 30–40, and so on. This would reduce the accuracy of the dataset, but it would still preserve some information about the people’s ages. On the other hand, if we suppress data points from the dataset, we would be losing information altogether.Perturbation involves modifying a feature’s value by replacing it with another value to minimize the difference between the original and modified datasets. One of the most popular perturbation techniques is microaggregation, which clusters records in the dataset with a restriction on cluster size. Microaggregation replaces the values of each record in the anonymized dataset with the values of the cluster center to which the record belongs. This approach is more effective than generalization and suppression [10] and simplifies further data analysis.

Table 1 shows a sample of de-identified original data, and Table 2 shows its 3-anonymized version. The Weight, Gender, and Age attributes are set as quasi-identifier attributes, and disease is the sensitive attribute. All records in a cluster share the same values for the quasi-identifier attributes, making them indistinguishable from each other. However, the 3-anonymity model in Table 2 does not prevent attribute disclosure. For example, if an adversary knows that a person’s record is in cluster C1 in Table 3, they can infer that the person has pneumonia.

While KA is effective in preventing identity disclosure, it remains vulnerable to attribute disclosure attacks. Specifically, if the range of sensitive attribute values within a k-anonymous cluster is limited, there is a risk of inferring the value of individuals’ sensitive attributes. Several extended KA models have been proposed to address this issue [11], including LD [6], which ensures a minimum degree of diversity (*l*) of sensitive values in each cluster.

Table 3 shows 3-anonymity and 2-diversity to the original data presented in Table 1. All records within each cluster in Table 3 have the same value for the quasi-identifier but contain distinct values for the sensitive attribute. This means that even if an adversary knows the cluster containing a target individual’s record, they cannot narrow down or disclose the person’s specific sensitive value (disease).

However, LD also has drawbacks [12]. For example, if an attacker knows that a person’s record is in cluster C2 in Table 3, they can infer that the person has the Flu with a probability of 66%. To address these limitations, Li et al. [7] introduced the concept of TC.

To meet TC requirements, the distribution of sensitive information must be similar for each group of indistinguishable records and the entire original dataset. Table 4 shows an anonymized version of the original data from Table 1 that satisfies 3-anonymity and 0.33-closeness. This anonymized table is safe for public release because it protects the privacy of the individuals in the dataset. In addition to protecting against the attacks mentioned above, TC also offers safeguards against potential skewness attacks and similarity attacks, as detailed in [7].

TC provides stronger privacy guarantees compared to other techniques like LD. It protects against attribute disclosure by restricting the distance between sensitive attribute distributions in any cluster and the original distribution. The parameter *t* effectively controls the balance between privacy protection and data utility. Decreasing *t* enhances privacy by requiring sensitive attribute distributions to more closely mirror the original distribution. However, it can reduce utility if overly strict clustering is needed to satisfy the tighter threshold. Conversely, increasing *t* may compromise privacy protection as it allows more divergence from the original distribution. However, it can improve utility by permitting more flexible clustering.

The Earth Mover’s Distance (EMD) [13] is used to calculate the TC metric using variational distance [7,14] as follows:(1)$$EMD\left(P,Q\right)=\frac{1}{2}\sum_{i=1}^{m}\left|p_{i}-q_{i}\right|$$
where $p_{i}$ and $q_{i}$ are the probabilities of distributions *P* and *Q*, respectively. It is assumed that |P| = |Q| = *m* and both *P* and *Q* are normalized distributions, that is $\sum_{i=1}^{m}p_{i}$ = $\sum_{i=1}^{m}q_{i}$ = 1.

As Figure 1 shows, there is an inverse relationship between data privacy and data utility, meaning that improving one inevitably reduces the other [15]. Therefore, a fundamental goal of anonymization is to balance these two factors when creating an anonymized dataset suitable for publication. This balance requires determining an acceptable trade-off between privacy and utility.

Table 5 summarizes the key abbreviations used throughout this paper for ease of reference.

### The Contribution

In recent years, researchers have explored clustering-based anonymization methods to protect people’s privacy from potential intruders [16,17,18,19,20]. These methods involve identifying important attributes to anonymize and using clustering techniques to group records according to their characteristics and similarities. The desired level of privacy protection can be achieved by modifying the attribute values of records within each cluster [21].

Byun et al. formulated the KA problem as a clustering problem and proposed the *k*-member algorithm to achieve optimal KA [20]. To enhance this algorithm, Honda et al. [22] developed a fuzzy variant of the *k*-member algorithm. Rahimi et al. [23] successfully implemented all three KA, LD, and TC restrictions on the original microdata using the X-means clustering algorithm. Langari et al. [24] also satisfied all three aforementioned constraints by formulating the anonymization problem as an optimization problem and subsequently employing an evolutionary algorithm and modifications to sensitive attribute values. Abbasi et al. [16] introduced a clustering process using the K-means++ method to achieve optimal KA.

Most real-world datasets contain non-numerical data (categorical data), which presents challenges for anonymization using generalization and suppression since categorical features lack natural distance metrics. For example, quantifying the distance between two different occupations, such as ‘dentist’ and ‘farmer’, is difficult. Therefore, most existing anonymization methods often focus on numerical data. This paper designs a novel approach for anonymizing categorical data using a greedy and information-theoretic clustering-based algorithm to protect microdata privacy against identity and attribute disclosure attacks.

The algorithm aims to minimize mutual information loss of records within each cluster while ensuring strict privacy protection. To this end, the clustering-based anonymization algorithm operates on a binary data table converted from the original data table. The output is an anonymized binary data table where some attribute values have been modified to achieve strict privacy protection.

The proposed method is specifically designed to anonymize non-numerical microdata so it can be publicly released for research and knowledge discovery while protecting privacy. We evaluated the proposed algorithm against four different approaches, including state-of-the-art methods, on three different datasets. The results demonstrated its superiority in achieving strict privacy protection while preserving data utility. It ensures that each cluster satisfies both KA and TC, protecting the anonymized data against identity and attribute disclosure attacks.

To summarize, the key contributions of this study are:Proposing a 3-phase anonymization algorithm that aims to achieve strict privacy protection while maximizing the data utility.To form clusters, we used the loss of mutual information and proposed an agglomerative clustering based on information bottleneck.Considering the diversity of sensitive values in addition to the similarity of quasi-identifier values during clustering to reduce the cost of implementing privacy constraints.

The paper is structured as follows: Section 2 introduces the key fundamental concepts necessary to understand privacy-preserving data anonymization, while Section 3 reviews existing methods. Section 4 then provides a detailed explanation of the proposed anonymization algorithm. In Section 5, an empirical performance evaluation and comparison of GCAA is conducted against four state-of-the-art benchmark methods. Both the privacy protections and utility preservation of the different algorithms are analyzed and discussed. Finally, Section 6 offers conclusions and outlines potential future work.

## 2. Basic Definition

Datasets can be modeled as tables, with each row containing a record of an individual and each column containing values related to their features. Features in tables to be anonymized are often classified into four main categories [25].

*Direct identifiers*: Such as names, national ID numbers, and email addresses, allow for direct identification of individuals. Thus, it is essential to remove them from the original data table before anonymizing the data.*Quasi-identifiers*: They are attributes that do not directly identify an individual but can be used to do so when linked to external data sources.*Sensitive Attributes*: They are confidential information that individuals may not want to share publicly. Disclosing this information to third parties can be harmful and may be misused. Examples of sensitive attributes are disease, income, and political or religious views.*Non-Sensitive Attributes*: They are any information that does not fall into the other categories. Examples of non-sensitive attributes include physical characteristics such as eye or hair color. Non-sensitive attributes are only released if they are essential for further analysis.

Orooji et al. [26] classified disclosure into two types: identity disclosure and attribute disclosure.

*Identity Disclosure*: Individuals can be identified from a dataset with accurate identity disclosure. External information sources, such as voter lists, can be used to identify a person, as shown in Figure 2.*Attribute Disclosure*: It occurs when an attacker can link a person to a sensitive attribute in a dataset. This is often the result of identity disclosure, which uniquely identifies an individual and makes it easier to associate their sensitive attributes [27].

By linking a person to a record in the anonymized dataset using external sources, an attacker can identify that record and associate the values of its sensitive attributes with the person’s identity, a process known as re-identification. The KA is proposed to avoid re-identification.

### 2.1. k-Anonymity

Given a data table *T*, the KA model requires that every record in the modified data table $T^{*}$ be indistinguishable from at least *k*-1 other records with respect to their quasi-identifiers. To achieve KA, the original dataset is partitioned into equivalence classes, each containing at least *k* members sharing the same quasi-identifier values. These equivalence classes can be referred to as QI-groups [28] or QI-clusters [29].

### 2.2. l-Diversity

The LD is a more robust privacy notion than KA, addressing some of the latter’s limitations. A cluster is *l*-diverse if it contains at least *l* distinct values of the sensitive attribute. A table is considered *l*-diverse if all its clusters satisfy the LD criterion. The LD principle ensures that each cluster has at least *l* distinct values for the sensitive attribute [7].

### 2.3. t-Closeness

The primary objective of the TC is to restrict the information an attacker can extract about sensitive attribute values within a particular cluster. It achieves this by requiring the distribution of sensitive attribute values within each cluster to be sufficiently similar to the distribution of sensitive attribute values in the entire dataset. The TC privacy metric is successful when all clusters satisfy the constraint. A cluster satisfies TC if the distance between the two distributions is smaller than *t*.

### 2.4. Information Theory Concept

Because our new privacy framework uses an information theory metric, we briefly overview the basic concepts of information theory [30] used in this paper.

*Information Entropy*: Information entropy, also known as Shannon entropy, is the average amount of information in each event. The more uncertain an event is, the higher its entropy; the less uncertain an event is, the lower its entropy. Let *X* be a discrete random variable that takes its values from the set $X^{\prime }$. If $P(x_{i})$ is the probability density function of values $x_{i}$ ($x_{i}$∈$X^{\prime }$), the entropy of variable *X* is defined as follows:
(2)$$H\left(X\right)=-\sum_{x_{i}\in X^{\prime }}^{}P\left(x_{i}\right)\log P\left(x_{i}\right)$$

*Conditional Entropy*: It measures the uncertainty associated with a random variable given the value of another random variable. Let *Y* be another discrete random variable with values $y_{j}$ taken from the set $Y^{\prime }$. The conditional entropy H(Y|X) is calculated as follows:
(3)$$H\left(Y|X\right)=-\sum_{x_{i}\in X^{\prime }}P\left(x_{i}\right)\sum_{y_{i}\in Y^{\prime }}P\left(y_{i}|x_{i}\right)\log P\left(y_{i}|x_{i}\right)$$Conditional entropy measures the additional information required to predict the value of *Y* given the value of *X*, beyond what can be predicted from *X* alone.

As shown in Figure 3, it can be used to measure the uncertainty in predicting the value of a discrete random variable when the value of another variable is known. Mutual information measures the extent to which knowing the value of one variable enables the prediction of another.

*Mutual information*: Mutual information is expressed as follows:
(4)I(X;Y)=H(X)−H(X|Y)=H(Y)−H(Y|X)=I(Y;X)
where I(X;Y)=0 if and only if *X* and *Y* are independent random variables.

This concept enables us to identify which clusters can be merged at each stage of the clustering process to minimize mutual information loss.

*Kolbeck-Leibler (KL) divergence*: KL divergence, also known as relative entropy, is a metric that measures the dissimilarity between two probability distributions. It provides a criterion for determining how different one distribution is from another. A KL divergence value of 0 indicates that the two distributions are similar in behavior, though not necessarily identical. Conversely, a value of 1 for this measure implies that the two distributions exhibit opposite behaviors. Given two probability distributions, *P* and *Q*, defined over a set *F*, the KL divergence quantifies how much they diverge from each other.
(5)$$D_{KL}\left[P||Q\right]=\sum_{f\in F}^{}P\left(f\right)\log\frac{P(f)}{Q(f)}$$

## 3. Related Work

Most state-of-the-art anonymization techniques use a clustering-based KA algorithm to achieve privacy preservation. These techniques partition input data into distinct clusters, containing a minimum of *k* records, before applying anonymization operations to create an anonymized output. The following section investigates several representative approaches to privacy-preserving.

LeFevre et al. [31] introduced the *Incognito* algorithm, a bottom-up approach that utilizes global generalization to generate an optimal *k*-anonymous dataset. However, this method has a high level of IL due to the full-domain generalization technique, which can negatively impact the usefulness of the anonymized data. To address this issue, Ref. [32] proposed a top-down greedy algorithm called *Mondrian*, which uses local recoding to generate an optimal *k*-anonymous dataset.

Byun et al. [20] proposed a clustering-based approach to solve the KA problem using a *k*-member greedy algorithm to find the optimal solution. The method partitions the dataset into clusters with at least *k* members and generalizes the quasi-identifier values in each cluster to achieve KA. However, it is sensitive to outlier records because it selects the furthest record from the previous cluster to create a new one, which leads to a high amount of IL. Honda et al. [22] introduced a fuzzy version of this approach that considers the degree of fuzzy membership when assigning records to clusters. However, this approach may overfit the data to satisfy KA, a significant drawback.

Abbasi et al. [16] proposed a clustering-based approach for privacy preservation in healthcare datasets. They first used the normal distribution function to identify and remove less frequent data, which is more vulnerable to privacy breaches. Then, they used the K-means++ algorithm to generate clusters during the clustering process. Finally, they applied the KA algorithm to each cluster separately to achieve the desired level of privacy preservation.

Truta et al. [29] proposed a *p*-sensitive *k*-anonymous technique to protect against homogeneity and background knowledge attacks. The algorithm ensures that each equivalent class has at least *p* distinct values (p≤k). However, this technique leads to a high degree of IL when the distribution of sensitive attribute values is not uniform. To enhance this approach, Sun et al. [28] proposed two improved versions of the *p*-sensitive *k*-anonymity technique: $p^{+}$-sensitive *k*-anonymity and (p,α)-sensitive *k*-anonymity. Their approach first categorizes the sensitive attributes and then focuses more on the categories of sensitive values rather than their specific values. However, this method may still result in a high amount of IL in particular scenarios.

Solanas et al. [33] proposed a heuristic microaggregation-based algorithm to apply *p*-sensitive *k*-anonymity to the original dataset. Their algorithm first creates clusters with at least *k* members that satisfy the *p*-sensitive *k*-anonymity property. To achieve this, it randomly selects a record and then adds the closest records with different sensitive values to form the cluster. Finally, the algorithm anonymizes the original dataset by replacing the values of each record with the values of the cluster centroid to which the record belongs. This method reduces the loss of information and provides higher data utility than other *p*-sensitive *k*-anonymity techniques.

Existing approaches that incorporate diversity in sensitive values have limitations, especially in protecting individuals’ privacy from attribute disclosure attacks. Researchers have suggested a privacy-preserving model known as TC to address this drawback.

Rahime et al. [23] proposed a 3-layer clustering algorithm that applied KA, LD, and TC constraints for anonymizing the original dataset. This approach protected individuals’ privacy against identity and attribute disclosure attacks, but its main weakness was the high amount of IL.

Langari et al. [24] proposed a hybrid approach for anonymizing graph and tabular data. Their method first partitions the data into clusters with at least *k* members using the fuzzy clustering technique and then uses the Firefly evolutionary algorithm to optimize the anonymization and clustering parameters simultaneously. LD and TC were implemented using the modification of sensitive values. However, its scalability is questionable due to the use of an evolutionary algorithm.

### Our Contributions against the Existing Methodology

Table 6 summarizes several state-of-the-art algorithms. These algorithms use different anonymization techniques and operations. According to the table, most algorithms were designed to satisfy only the KA constraint and do not adequately protect anonymized data against attribute disclosure attacks. Algorithms that provide more protection by satisfying LD or TC constraints suffer from a high amount of IL or low data utility. This is because almost all of these algorithms only consider the similarity of quasi-identifiers when creating clusters. Therefore, to satisfy the privacy limitation, they are forced to merge more clusters, which in the worst case, could result in a single cluster with zero data utility.

Most anonymization techniques use generalization operations, which tend to incur high levels of IL (except for the algorithms in [24,33]). However, anonymization inherently reduces data usefulness to some degree by obscuring details. Therefore, it is crucial to consider the trade-off between privacy and IL to maintain the utility of anonymized data for knowledge extraction.

Compared to the proposed method, most investigated approaches perform poorly in maintaining the trade-off between privacy protection and data usefulness. Additionally, unlike the proposed method, some of these techniques remove data points from the table as outliers, while others change the values of the sensitive attributes to meet the LD and TC requirements. This leads to a significant decrease in the usefulness of the data.

To overcome the limitations of prior work, a new greedy, information-theoretic clustering algorithm is proposed for microdata anonymization. It differs from existing techniques in several important ways:The proposed algorithm clusters data points by considering both the similarity of quasi-identifier values and the diversity of sensitive values within each cluster. This approach ensures that clusters have high intra-cluster similarity for quasi-identifiers and high intra-cluster diversity for sensitive attribute, which reduces both the modification of values and the merging of clusters as a result of increasing the usefulness of the data.The proposed algorithm protects released data from identity and attribute disclosure attacks by satisfying the KA and TC privacy constraints. These constraints ensure that individuals are indistinguishable from the anonymized data and that the distribution of sensitive attribute values is preserved.As shown in Figure 1, there is a trade-off between data privacy and data utility. Existing methods prioritize data privacy at the expense of data utility, but our algorithm achieves strict privacy by adhering to KA and TC while minimizing IL. This approach strikes a balance between data privacy and utility, ensuring that the anonymized data remains useful for its intended purposes.

## 4. Proposed Approach

Creating a dataset that satisfies the TC constraint through generalization involves solving an optimization problem to determine the minimum generalization required to meet the TC requirement. A common strategy for this is an iterative process of progressively generalizing one or more attributes until the resulting dataset meets the TC constraint.

Traditional methods for achieving TC first cluster records into groups containing at least *k* members to ensure KA. KA can be achieved through generalization techniques that replace quasi-identifier values with broader categories or microaggregation techniques that substitute original values with statistical approximations. Next, the methods merge clusters of records to construct a *t*-close dataset. The clusters are progressively merged to minimize the distance between sensitive attribute distributions across merged clusters. The distance metric guides the selection of cluster mergers, identifying combinations that most reduce divergence between clusters’ sensitive field distributions. This process continues by merging clusters until all satisfy the threshold for TC.

Generating *t*-close datasets using the traditional methods is costly, primarily due to the reduction in data utility. A key factor is the lack of consideration for sensitive attribute value diversity during initial clustering. Real-world datasets often contain records with similar quasi-identifiers but different sensitive attribute values. To enhance data utility, our proposed approach generates clusters having more similar quasi-identifiers and more diverse sensitive values. Additionally, merging clusters to achieve *t*-closeness can severely compromise data utility, sometimes rendering it entirely useless (all clusters merged into one). A more effective strategy could be to swap records between clusters instead of merging clusters, which could potentially mitigate this problem.

Pre-processing, clustering, and adjustment are the three phases of the proposed approach to generating anonymized datasets. In the pre-processing phase, the original data table *T* is converted into a binary table, $T_{b}$, with values of 1 or 0. $T_{b}$ is then normalized to the range [0, 1] and divided into two subtables: quasi-identifier, $T_{QI}$, and sensitive attribute, $T_{SA}$. In the clustering phase, a bottom-up clustering algorithm is used to generate clusters with similar quasi-identifier values but distinct sensitive values within each cluster. Finally, to create the anonymized table $T^{*}$, which satisfies the KA and TC requirements for strict privacy, we apply an adjustment process. The quasi-identifier values in each cluster are replaced with the centroid quasi-identifier value to anonymize the data. Figure 4 shows the overall workflow of the proposed approach.

### 4.1. Pre-Processing

The original input data table *T* contains *n* records from the set R, each of them having *m* values of attributes from the set F which is represented by the matrix *M*.

First of all, attribute values are converted to binary values. Let feature $F_{j}$ from the set F have a value domain $F_{j}$ = {$v_{j1}$, $v_{j2}$,..., $v_{jd_{j}}$} (for each j=1,2,…,m). Therefore, each record r∈R takes exactly one value from the set $F_{j}$ for its *j*th feature. Let *F* = $F_{1}$∪$F_{2}$∪...∪$F_{m}$ be the set of all possible values of the attributes in the original table. Also, let $d=d_{1}+d_{2}+\cdots +d_{m}$ be equal to the size of *F*.

In this case, the original matrix converted to a matrix $M_{b}$ with the size n×d, where each record r∈R in this matrix is a vector with binary values of features size *d*. If the record *r* contains the attribute value *f*, the value of $M_{b}\left[r,f\right]$ equals 1, otherwise is equal to 0. For example, according to Table 7, the first feature is the *Gender*, and its values domain is {*Male*, *Female*}. In the binary table, if the person is male, only the attribute *Male* value is 1 and *Female* value is 0, and vice versa.

Next, we identify quasi-identifiers and sensitive attributes in the data table and partition the feature set F into two sets, $F_{\mathrm{QI}}$ and $F_{\mathrm{SA}}$. We then construct two sub-tables, $T_{QI}$ and $T_{SA}$, from the original data table, ensuring that each record retains its unique ID across both sub-tables. In the next step, we normalize the values in both sub-tables to lie within the interval [0, 1], so that the sum of the values for each record in each sub-table equals 1.

Let *R* and *F* ($F_{QI}$ and $F_{SA}$) be discrete random variables whose values are taken from sets R and F, respectively. In this case, each record in the two sub-tables has the conditional probability distribution $P(f_{QI}|r)$ (for all $f_{QI}$∈$F_{QI}$) and $P(f_{SA}|r)$ (for all $f_{SA}$∈$F_{SA}$), respectively. The normalized sub-tables of the original Table 7 are shown in Table 8 and Table 9.

### 4.2. Clustering

The two sub-tables generated in the previous phase, $T_{QI}$ and $T_{SA}$, serve as input to the clustering phase. The proposed bottom-up clustering algorithm simultaneously considers both $T_{QI}$ and $T_{SA}$ to group the records. The goal is to form clusters where quasi-identifier values are highly similar within each cluster, but sensitive values remain diverse. Algorithm 1 outlines the bottom-up clustering process.
**Algorithm 1** The Pseudocode for the Clustering Process1:**Input:** Microdata table *T* with n×m dimensions, parameters *K*, *T*, β, and γ2:**Output:** Clusters $C_{1},C_{2},\ldots ,C_{c}$ (c<n)3:Let *n*: the number of records, *m*: be the number of features, and *d*: the size of the feature values domain, *S*: the distribution of sensitive values in the dataset.4:Convert data table *T* to binary table $T_{b}$ with n×d dimensions5:Select quasi-identifiers and sensitive attributes, then establish two sub-tables $T_{QI}$ and $T_{SA}$ with unique ID6:Initially, let each record be a distinct cluster, $C_{n}=\left(c_{1},c_{2},\ldots ,c_{n}\right)$7:Calculate $P(c_{i})$ based on Equation (Equation 6) for i=1,2,…,n8:**while** $\delta I\left(c_{n+1}\right)-\delta I\left(c_{n}\right)\leq \gamma$ **do**9:     Calculate $\delta I(c_{i},c_{j})$ based on Equation (Equation 13) for all i≠j10:    Select a pair of clusters $c_{x}$, $c_{y}$ that have minimum $\delta I(c_{x},c_{y})$11:    **if** $|c_{x}\left|+\right|c_{y}|\leq K+\beta$ **then**12:        **if** $EMD\left(c_{x}\cup c_{y},S\right)<\min\left(EMD\left(c_{x},S\right),EMD\left(c_{y},S\right)\right)$ **then**13:           Merge $c_{x}$ and $c_{y}$, and update both sub-tables based on Equations (Equation 8) and (Equation 9) and go to Line 714:        **else**15:           Select the next pair of clusters and go to Line 1216:        **end if**17:    **else**18:        Select the next pair of clusters and go to Line 1119:    **end if**20:**end while**21:Return obtained clusters $C_{1},C_{2},\ldots ,C_{c}$

A *k*-clustering $C_{k}$ partitions the records of data table *T* into *k* clusters $C_{k}=${$c_{1},c_{2},c_{3},\ldots ,c_{k}$}, where each cluster $c_{i}\in C_{k}$ has three properties as follows: (1) $|c_{i}|\neq 0$, (2) $c_{i}\cap c_{i}=0$ for all i,j,i≠j, (3) $\cup_{i=1}^{k}c_{i}=T$. We define the probability of each record *r* in data table *T* as p(r)=1/n, where *n* is the number of records in *T*. Then, for each cluster $c\in c_{k}$ we have:(6)$$P\left(c\right)=\sum_{r\in c}P\left(r\right)=\frac{\left|c\right|}{n}$$

For example, from Table 8 and Table 9, one can conclude that $P\left(r_{i}\right)=\frac{1}{5}$ for i=1,2,…,5. In addition, $P(f_{QI}|r_{1})$ and $P(f_{SA}|r_{i})$ are the feature vectors for the quasi-identifier and sensitive attribute of record $r_{1}$, respectively. They are equal to [1/3,0,1/3,0,1/3,0,0] and [1/2,0,0,1/2].

Initially, each record is assumed to be an independent cluster. We denote this partition as $C_{n}$, where $C_{n}$ is a partition with *n* clusters {$c_{1}$, $c_{2}$,..., $c_{n}$} such that:(7)$$P\left(c_{i}\right)=1/n,\left(i=1,2,\ldots ,n\right)$$

At each iteration, the proposed algorithm seeks to generate a new cluster with similar quasi-identifier values and distinct sensitive values from the existing clusters. To this end, we employ the Agglomerative Information Bottleneck algorithm [34]. The algorithm starts with a $C_{n}$ partition. In this case, I(F;R) = $I(F;C_{n})$, and in each step, it merges two clusters, reducing the number of clusters by one. According to the definition of mutual information, this causes an increase in the uncertainty of predicting the features with respect to clusters, or, equivalently, a reduction of the information that clusters contain about the values of the features, i.e., $I\left(F;C_{n-1}\right)\leq I\left(F;C_{n}\right)$. We use δI to represent this information reduction. Therefore, at each step, the algorithm attempts to merge the pair of clusters with the minimum δI.

After merging two clusters $c_{i}$ and $c_{j}$, new cluster $c_{ij}$ has:(8)$$P\left(c_{ij}\right)=P\left(c_{i}\right)+P\left(c_{j}\right)$$
(9)$$P\left(f|c_{ij}\right)=\frac{P(c_{i})}{P(c_{ij})}P\left(f|c_{i}\right)+\frac{P(c_{j})}{P(c_{ij})}P\left(f|c_{j}\right)$$
for all *f*∈*F*. Tishaby et al. [35] showed that:(10)$$\delta I\left(c_{i},c_{j}\right)=P\left(c_{ij}\right)\times JSD\left[P\left(f|c_{i}\right),P\left(f|c_{j}\right)\right]$$
for all f∈F. JSD in Equation (Equation 10) is the Jensen-Shannon divergence defined below:(11)$$JSD\left[P\left(f|c_{i}\right),P\left(f|c_{j}\right)\right]=\frac{P(c_{i})}{P(c_{ij})}D_{KL}[P\left(f|c_{i}\right)\left|\right|\bar{P}]+\frac{P(c_{j})}{P(c_{ij})}D_{KL}[P\left(f|c_{j}\right)\left|\right|\bar{P})]$$
where $D_{KL}$ is KL divergence, and $\bar{P}$ is avarge distribution of $P(f|c_{i})$ and $P(f|c_{j})$ and calculated by Equation (Equation 9). Finally, the δI$(c_{i}$, $c_{j})$ which i≠j is calculated with the following formula:(12)$$\delta I\left(c_{i},c_{j}\right)=P\left(c_{i}\right)\sum_{f\in F}P\left(f|c_{i}\right)\log\frac{P(f|c_{i})}{P(f|c_{ij})}+P\left(c_{j}\right)\sum_{f\in F}P\left(f|c_{j}\right)\log\frac{P(f|c_{j})}{P(f|c_{ij})}$$
where $\delta I(c_{i},c_{j})$ = $\delta I(c_{j},c_{i})$.

At each step, the proposed algorithm calculates δI of each pair of clusters $c_{i},c_{j}$ (i≠j) for both sub-tables $T_{QI}$ and $T_{SA}$ as $\delta I_{QI}\left(c_{i},c_{j}\right)$ and $\delta I_{SA}\left(c_{i},c_{j}\right)$, respectively. Then $\delta I_{F}\left(c_{i},c_{j}\right)$ is calculated according to the following formula:(13)$$\delta I_{F}\left(c_{i},c_{j}\right)=\delta I_{QI}\left(c_{i},c_{j}\right)-\delta I_{SA}\left(c_{i},c_{j}\right)$$

At each step, the algorithm aims to merge the pair of clusters with the minimum $\delta I_{F}$ value. However, if two clusters have identical quasi-identifiers (i.e., $\delta I_{QI}=0$), the algorithm selects the pair of clusters with the highest diversity in sensitive values for merging. By following this strategy, we can step-by-step create clusters of records with similar quasi-identifiers but different sensitive values. This process reduces IL and improves the quality of anonymous data.

To create balanced clusters and minimize changes during TC enforcement, our algorithm imposes two additional restrictions on merging. To speed up the process and create more evenly sized clusters, we remove any cluster that has between *k* and k+β (i.e., k≤|c|≤k+β) data points from the clustering process. To make it easier to enforce TC during the adjustment phase, two clusters are merged only if they meet the following criteria:(14)$$EMD\left(S_{c_{ij}},S\right)<min\left(EMD\left(S_{c_{i}},S\right),EMD\left(S_{c_{j}},S\right)\right)$$
where *S* is the distribution of sensitive values in the entire dataset, and $S_{c_{i}}$ is the distribution of sensitive values in cluster *i*.

The algorithm runs until it reaches a stopping point, defined as follows: if the change in mutual information after each merging step $\delta I_{F}$ exceeds the threshold γ, the merging process stops, and the final clusters are formed.

### 4.3. Adjustment

The adjustment phase is a crucial step in the proposed algorithm to satisfy the KA and TC constraints. It takes as input the set of clusters generated in the clustering phase, denoted as $C_{1}$, $C_{2}$,..., $C_{c}$, and outputs a new set of clusters, denoted as $C_{1}^{*}$, $C_{2}^{*}$,..., $C_{c^{*}}^{*}$ ($c^{*}\leq c$), that all satisfy the KA and TC constraints while preserving the data quality. The resulting anonymized table $T^{*}$ can be easily constructed using the modified clusters. Algorithms 2 and 3 present the pseudocode for the adjustment phase.
**Algorithm 2** The Pseudocode for the Adjustment Phase1:**Input:** Clusters $C_{1},C_{2},\ldots ,C_{c}$, parameter *K* and *T*2:**Output:** Clusters C${}_{1}^{*},$ C${}_{2}^{*},\ldots ,$C${}_{c^{*}}^{*}$ ($c^{*}\leq c$), Anonymized table $T^{*}$3:Let $D_{1}$, $D_{2}$, $D_{21}$, and $D_{22}$ be empty sets4:Let SortedList: Sort all clusters based on their *EMD* in descending order5:**for** i = 1, 2,..., c **do**6:    **if** $C_{i}$ satisfy KA and TC **then**7:        ADD $C_{i}$ To $D_{1}$8:    **else**9:        ADD $C_{i}$ To $D_{2}$10:    **end if**11:**end for**12:**for** j = 1, 2,..., $|D_{2}|$ **do**13:    **if** $|C_{j}|$ ≥ *K* **then**14:        ADD $C_{j}$ To $D_{21}$15:    **else**16:        ADD $C_{j}$ To $D_{22}$17:    **end if**18:**end for**19:**for** all clusters *c* in $D_{22}$ **do**20:    **while** *c* is not empty **do**21:        Randomly select a record *r* From *c*22:        *c* = *c*∖{r}23:        *p* = ***Findthebestcluster***(SortedList, *r*)24:        *p* = *p*∪{r}25:    **end while**26:**end for**27:**for** all clusters *c* in $D_{21}$ **do**28:    **while** *c* is not empty **do**29:        Randomly select a record *r* from *c*30:        *c* = *c*∖{r}31:        *p* = ***Findthebestcluster***(SortedList, *r*)32:        *p* = *p*∪{r}33:    **end while**34:**end for**

**Algorithm 3** Function: Findthebestcluster(C,r)
1:**Input:** A set of clusters C that sorted based on their *EMD* in descending order, and a record r2:**Output:** A cluster *c*∈*C* such that the EMD(*c*∪{r}, *S*) is decreased3:Let *n* = |C|, best = null4:**for** *i* = 1, 2,..., *n* **do**5:    **if** EMD($c_{i}$∪*r*, *S*) <EMD($c_{i}$, *S*) **then**6:        best = $c_{i}$ with the lowest ADR value7:        break8:    **end if**9:
**end for**
10:Return best


The input set of clusters is partitioned into two subsets: $D_{1}$, which contains the clusters that satisfy both KA and TC, and $D_{2}$, which contains the clusters that do not meet one or both of these constraints. $D_{2}$ is further partitioned into two subsets: $D_{21}$, which contains clusters with more than *k* records, and $D_{22}$, which contains clusters with less than or equal to *k* records.

The algorithm then proceeds to modify the clusters in $D_{22}$ by transferring their records to clusters in $D_{21}$ using the ***Findthebestcluster*** function to minimize the EMD between the merged clusters. If the EMD of the modified cluster is less than or equal to the predefined threshold *T*, the cluster is moved from $D_{21}$ to $D_{1}$.

The process continues until $D_{22}$ is empty. If $D_{21}$ is not empty, the algorithm selects the cluster with the highest EMD value and transfers its records to the best cluster based on the ***Findthebestcluster*** function until all clusters in $D_{21}$ satisfy the TC constraint and are moved to $D_{1}$.

## 5. Performance Evaluation and Discussion

To evaluate the proposed algorithm, we use two metrics to measure the quality of anonymized datasets.

Normalized Avarege QI-group (CAVG): The metric was proposed by [32]. It measures the quality of the KA constraint. The closer the value to 1, the better the result. The normalized average group is defined as follows:
(15)$$CAVG=\frac{N}{C\times k}$$
where C is the number of clusters and *k* is the privacy parameter.Average Distortion Ratio (ADR): This metric measures the amount of distortion (IL) introduced by the anonymization process. If a feature value is not changed, there is no distortion. If a feature value is changed, the distortion increases by 1. The metric is defined as follows:
(16)$$ADR=\frac{\sum_{i=1}^{N}\sum_{j=1}^{d}F_{ij}}{N\times d}$$
where $F_{ij}=1$ if *j*th features of record *i* is modified and 0 otherwise, *N* is the number of records, and *d* is the domain of the feature values.

To assess the effectiveness of the algorithm in protecting against identity and attribute disclosure attacks, we use two penalty criteria: PKA and PTC. These criteria are defined as follows:P${}_{KA}$ is incremented by 1 if any cluster does not satisfy the KA constraint.P${}_{TC}$ is incremented by 1 if any cluster does not satisfy the TC constraint.

These criteria enable an evaluation of the algorithm’s ability to satisfy the desired privacy constraints while minimizing the risk of information disclosure.

The proposed anonymization algorithm was developed in Python version 3 and implemented on a personal computer with a 2.6 GHz (Core i7) processor and 16 GB of memory. The effectiveness of the proposed Greedy Clustering-based Anonymization Algorithm (GCAA) was compared to four other clustering-based anonymization techniques: K-anonymity L-diversity T-closeness 3Layer (KLT3L) [23], K-member Fuzzy Clustering Firefly Algorithm (KFCFA) [24], P-Sensitive K-Anonymity (PSKA) [28], and Fuzzy K-Member Clustering (FKMC) [22]. The comparison was performed in terms of their ability to protect against identity and attribute disclosure attacks.

To ensure a fair comparison with existing methods, we designed our experiment to match the setup of [24] as closely as possible. We used three microdata sets (https://snap.stanford.edu/data/, accessed on 10 February 2023) that were also used in the [24]. Each dataset contained a table of records, with each record having a set of features. The details of the microdata sets are summarized in Table 10.

The level of protection against intruder attacks increases with higher values of *k* and *t* but at the cost of increased I. To assess the performance of our algorithm under different conditions, we considered three different privacy scenarios, each with different parameter values. A summary of the scenarios is presented in Table 11.

Considering the privacy requirements and the potential severity of intruder attacks, we chose different values for the anonymization parameters *k* and *t* for each privacy scenario. As shown in Table 11, we considered three scenarios, ranging from simple to strict, with corresponding values of *l* set to 3, 4, and 6, respectively. We also set the parameters β and γ to ⌊K/2⌋ and 0.5, respectively. To terminate the clustering process, we monitored the difference between two consecutive δI values for merging. Clustering was stopped when this value exceeded 0.5, at which point the current clusters were considered to be the final clusters.

### 5.1. Experimental Result for Simple Privacy Scenario

Table 12 displays the results obtained for a simple privacy scenario on the Facebook micro dataset using the proposed algorithm. In this scenario, the algorithm generated 70 clusters from the data. Cluster sizes ranged from a minimum of four records to a maximum of eight records. All clusters satisfied the LD requirement, containing at least *l* distinct sensitive attribute values. Sensitive attribute diversity within clusters varied between 3 and 7 different values.

Importantly, all clusters generated by the algorithm satisfied both KA and TC privacy criteria. This is because the clustering process prioritized forming groups with a high diversity of sensitive values. Additionally, none of the 70 clusters produced were susceptible to attribute disclosure attacks. By ensuring sensitive attribute diversity within each cluster, the proposed method successfully anonymized the data while preventing homogeneity and skewness of sensitive values that could otherwise compromise privacy according to the attribute disclosure attack model.

It is worth noting that, unlike other techniques, our proposed approach did not remove any records from the original data table as outliers. These results demonstrate the effectiveness of the algorithm in achieving both KA and TC privacy goals while preserving data utility, as shown by retaining all records. A key advantage of our approach compared to other anonymization methods is the ability to satisfy strong privacy standards without eliminating any records. By clustering records rather than discarding outliers, more of the original information can be retained for analysis purposes. Maintaining all records has important implications for data utility, as no information is lost due to outlier removal. This highlights the strengths of the proposed method.

Table 12 shows that while the FKMC method performed best in terms of CAVG and ADR, it suffered from a significantly high P${}_{TC}$ error. Although PSKA and FKMC had no P${}_{KA}$ errors, they remained susceptible to attribute disclosure due to their non-zero P${}_{TC}$ penalty errors. Only the KLT3L and KFCFA methods generated clusters that fully satisfied the TC constraint, making them truly comparable to the proposed GCAA approach in terms of privacy protection.

The proposed GCAA method generates more balanced clusters than KLT3L and KFCFA, which improves CAVG by 78% and 33%, respectively, compared to KLT3L and KFCFA. Moreover, GCAA significantly reduces ADR compared to KLT3L, although it does not outperform KFCFA. KFCFA changes sensitive values to achieve TC, which significantly reduces data utility, whereas GCAA does not require such changes.

Table 12 also reports the average CPU running time of each model. The proposed GCAA algorithm showed computational performance that was nearly three times faster than the KFCFA approach. This significant difference can be attributed to KFCFA’s use of an evolutionary algorithm during its optimization process, which induces greater computational overhead compared to GCAA.

Figure 5, Figure 6 and Figure 7 show a comparative analysis of C, CAVG, and ADR for three microdata sets. The results show that the proposed GCAA algorithm performs better than KLT3L on all three datasets and outperforms KFCFA in most cases.

Figure 5 reveals that GCAA produces more balanced clusters than KLT3L for all three datasets, resulting in higher C values. Moreover, Figure 6 shows that GCAA consistently outperforms KLT3L in terms of CAVG across all datasets. Finally, Figure 7 demonstrates that GCAA significantly reduces ADR compared to KLT3L on all three datasets.

In addition, GCAA achieves competitive results with KFCFA, outperforming it in terms of C and CAVG on two datasets, as shown in Figure 5 and Figure 6. While KFCFA performs slightly better than GCAA in terms of ADR on one dataset, Figure 7 shows that GCAA still achieves excellent results with a lower ADR than KFCFA on the other two datasets.

### 5.2. Experimental Result for Medium Privacy Scenario

To limit the anonymization process, the experimenters increased *k* to 6 and decreased *t* to 0.5. The results were compared across different models, as shown in Table 13.

Similar to the previous scenario, FKMC outperformed other methods in terms of C and CAVG. At the same time, GCAA, KLT3L, and KFCFA were superior to PSKA and FKMC due to more robust privacy protection. These methods satisfied the TC constraint in all clusters, resulting in zero P${}_{TC}$ error. KLT3L merged many clusters to satisfy the TC constraint, resulting in a reduction in the number of final clusters to 12. This resulted in a larger CAVG of 4.82, which is almost four times larger than GCAA, and an ADR that was twice as large. KFCFA outperformed the proposed GCAA in this experiment, as shown in Figure 8 and Figure 9.

These figures compare the C and CAVG criteria for GCAA, KLT3L, and KFCFA in the medium privacy scenario on three microdata sets. The proposed GCAA had a lower CAVG error in all datasets than KLT3L. KFCFA performed better than GCAA on the Facebook and Twitter datasets, but GCAA had fewer errors than KFCFA on the Google+ dataset. Figure 10 shows a similar comparison for ADR criteria, with the proposed GCAA performing better than KLT3L in most cases.

### 5.3. Experimental Result for Strict Privacy Scenario

To evaluate the algorithms under an even more restrictive privacy setting, the experimenters increased the KA parameter *k* to 8 and decreased the TC parameter *t* to 0.3. Table 14 shows the results of different approaches on the Facebook microdata set. KLT3L and KFCFA generated 4 and 25 clusters, respectively, with CAVGs of 10.48 and 1.735. The proposed GCAA algorithm improved the ADR error by 70.5% and 0.7% compared to KLT3L and KFCFA, respectively.

Figure 11, Figure 12 and Figure 13 compare the C, CAVG, and ADR results obtained from the proposed GCAA, KLT3L, and KFCFA under the strict privacy scenario on three microdata sets. The proposed approach showed superior performance by generating more balanced clusters on the Twitter and Google+ datasets, resulting in less error than the other two methods for CAVG and ADR criteria. Although KFCFA produced more clusters on the Facebook dataset than the proposed GCAA, the ADR error was not lower because KFCFA significantly changed the feature values to achieve TC.

### 5.4. Experimental Result for Different Values of Parameter T

To investigate the impact of the parameter *t*, a set of experiments was carried out where the parameter *t* was systematically varied from 0.3 to 0.7, while holding *k* and *l* fixed at 6 and 4, respectively. In this analysis, the Google+ microdata set was used to observe the effect of changing *t* on the C, CAVG, and ADR criteria. Since decreasing *t* complicates the problem of achieving the TC constraint, it is expected to increase various error rates. As a result, with a decrease in *t*, CAVG increases, indicating the appearance of larger clusters.

Figure 14 and Figure 15 show that the proposed GCAA performs better than KLT3L and KFCFA in terms of C and CAVG metrics, especially in scenarios where privacy is very important. However, as seen in Figure 16, ADR increases for all methods as *t* decreases. Despite its shortcomings, the proposed GCAA still outperforms KLT3L and KFCFA in strict privacy settings.

### 5.5. Discussion

In terms of privacy, the proposed GCAA strategy along with KLT3L and KFCFA fully addresses the risks of identity and attribute disclosure attacks by satisfying both the KA and TC requirements. The PSKA and FKMC methods only partially mitigate these risks, as they do not perfectly satisfy TC. However, the resistance of GCAA and some other algorithms to similarity attacks could be improved. Specifically, they currently lack a criterion for determining the semantic similarity of attribute values during clustering. For example, in a medical context, the algorithms have no way to judge that diseases like colon cancer and liver cancer should be considered highly similar rather than diverse, for the purpose of TC. Not accounting for such attribute-relatedness weakens protections against similarity attacks.

In terms of data utility, FKMC and PSKA methods had the lowest amount of IL, as measured by ADR, and therefore the best data utility. The proposed method and KFCFA performed similarly well, but the KLT method lost a significant amount of IL. This suggests that the proposed method is effective at preserving the usefulness of anonymized data for future knowledge extraction.

In terms of computational efficiency, the clustering phase has a time complexity of $O(n^{2}\times d^{2}\times \log n)$, where *d* is the size of the attribute’s domain and *n* is the number of records. However, if n>>d, it can be concluded that the time complexity is $O(n^{2}\times \log n)$. The proposed adjustment phase for splitting and merging clusters has a time complexity of $O(n^{2})$.

In terms of average execution time, the proposed method outperforms KFCFA but underperforms other methods. KFCFA is slow, unscalable, and computationally expensive due to its evolutionary nature. The proposed method is also slower than other methods due to its time complexity. Because it provides a high level of privacy protection and data utility, the proposed method is suitable for anonymizing publishable data in inactive mode, where the data owner publishes the entire anonymized dataset after making some modifications to the original data [36]. In this case, the data owner prioritizes the level of privacy and utility of anonymized data.

These findings are consistent with the results of other experiments, highlighting the effectiveness of the proposed GCAA approach in achieving high privacy protection while minimizing the distortion of the original data. A brief review of the above, key findings emerge:As privacy protection increases, the data becomes more distorted, which reduces its usefulness. This is consistent with Figure 1.The proposed GCAA method, as well as the KLT3L and KFCFA methods, can protect data from identity and attribute disclosure attacks by meeting both KA and TC.The KLT3L algorithm has higher error measures (CAVG and ADR) than the proposed methods because it needs to merge more clusters and make more errors to meet the TC constraint.The proposed GCAA outperforms the KFCFA method in some cases, while the KFCFA method produces better results in other cases. However, changing the values of sensitive features to meet the TC constraint can reduce the usefulness of the data.The results demonstrate the effectiveness of the proposed GCAA algorithm. The main reasons are: (1) It considers the diversity of sensitive feature values and the similarity of quasi-identifier values simultaneously during clustering (2) It generates evenly sized clusters that are consistent with the KA and TC constraints. (3) It uses an adjustment algorithm to produce an anonymized dataset with maximum data utility.

## 6. Conclusions and Future Works

The proposed Greedy Clustering-based Anonymization Algorithm (GCAA) provides a novel contribution to the field of data privacy. It uses information theory concepts and performs bottom-up hierarchical clustering to form clusters with similar quasi-identifier values and diverse sensitive attribute values. The algorithm iteratively merges pairs of clusters with the minimum loss of mutual information and reduces the EMD criterion. Clusters with a cardinality between *k* and k+β removed from the clustering process to ensure that balanced clusters satisfying the KA and TC constraints are formed.

In the adjustment phase, the GCAA algorithm enforces the KA and TC constraints by splitting and merging clusters that do not meet these requirements while reducing the EMD. We evaluated GCAA’s performance based on the CAVG and ADR criteria and compared it with four other methods: KLT3L, KFCFA, PSKA, and FFKMC. Our experimental results demonstrate that GCAA efficiently forms balanced clusters while providing robust privacy protection and preserving data quality. GCAA is a faster method than KFCFA that provides similar levels of privacy protection and IL. However, the proposed algorithm may require a longer execution time than some other methods due to its time complexity.

The proposed algorithm exhibits several limitations. Firstly, it is tailored for non-numeric data, rendering it unsuitable for numerical data anonymization. Secondly, the algorithm is confined to single-sensitive attribute scenarios. Finally, the algorithm is restricted to tabular data and is ill-suited for graphical or structural data analysis. GCAA has practical applications in anonymizing scalable microdata where privacy preservation is crucial while retaining useful information for data mining purposes.

The primary objectives of our future research to enhance the proposed method include:Defining a criterion to evaluate the similarity between attribute values based on the proposed structure, thereby reinforcing the algorithm against similarity attacks.Extending the proposed method to handle datasets with multiple sensitive attributes.Expanding the algorithm to anonymize numerical data in addition to non-numerical data.Other suitable clustering methods for categorical data could also be considered, such as partitioning around medoids using the Gower coefficient [37]. This method is well-suited for datasets containing binary values, similar to the data used in our experiments. Evaluating the performance of partition around medoids and comparing it to the proposed approach may provide additional insights.Reducing the time complexity of the algorithm.

## Figures and Tables

**Figure 1 entropy-25-01613-f001:**
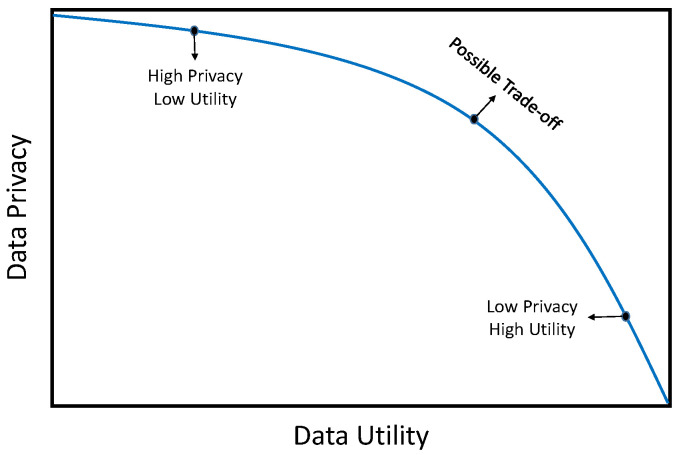
Trade-off between privacy and utility level.

**Figure 2 entropy-25-01613-f002:**
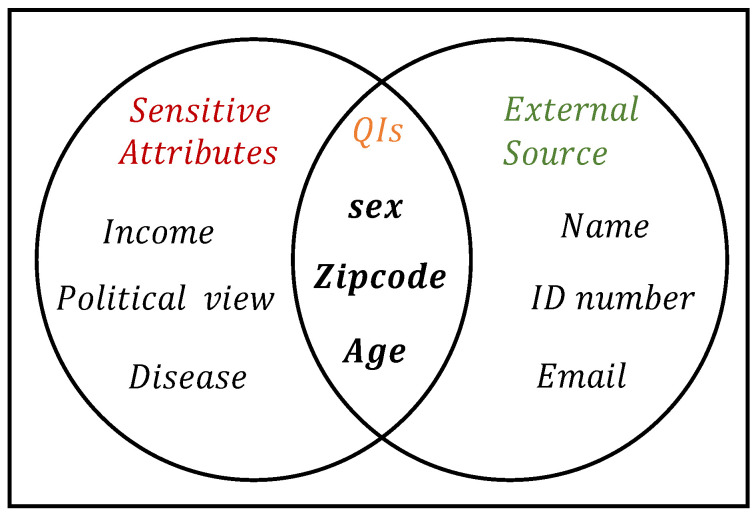
Identity disclosure using external information.

**Figure 3 entropy-25-01613-f003:**
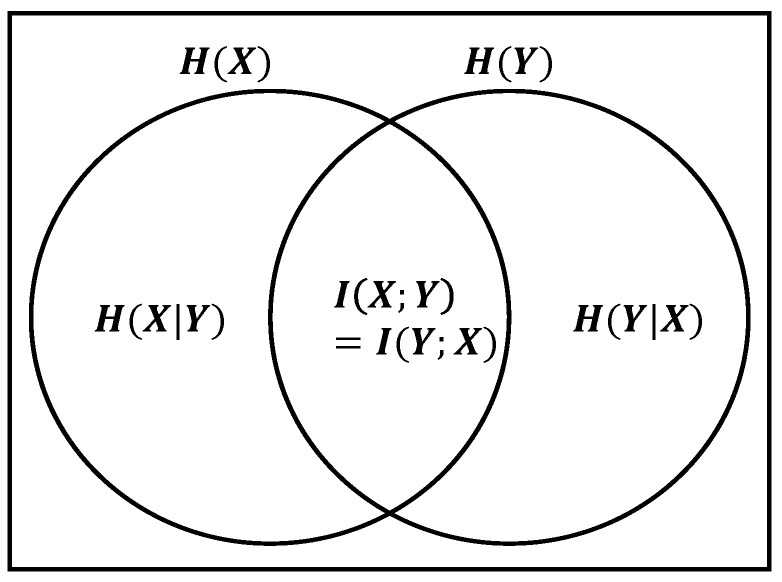
Relation between mutual information and conditional entropy.

**Figure 4 entropy-25-01613-f004:**
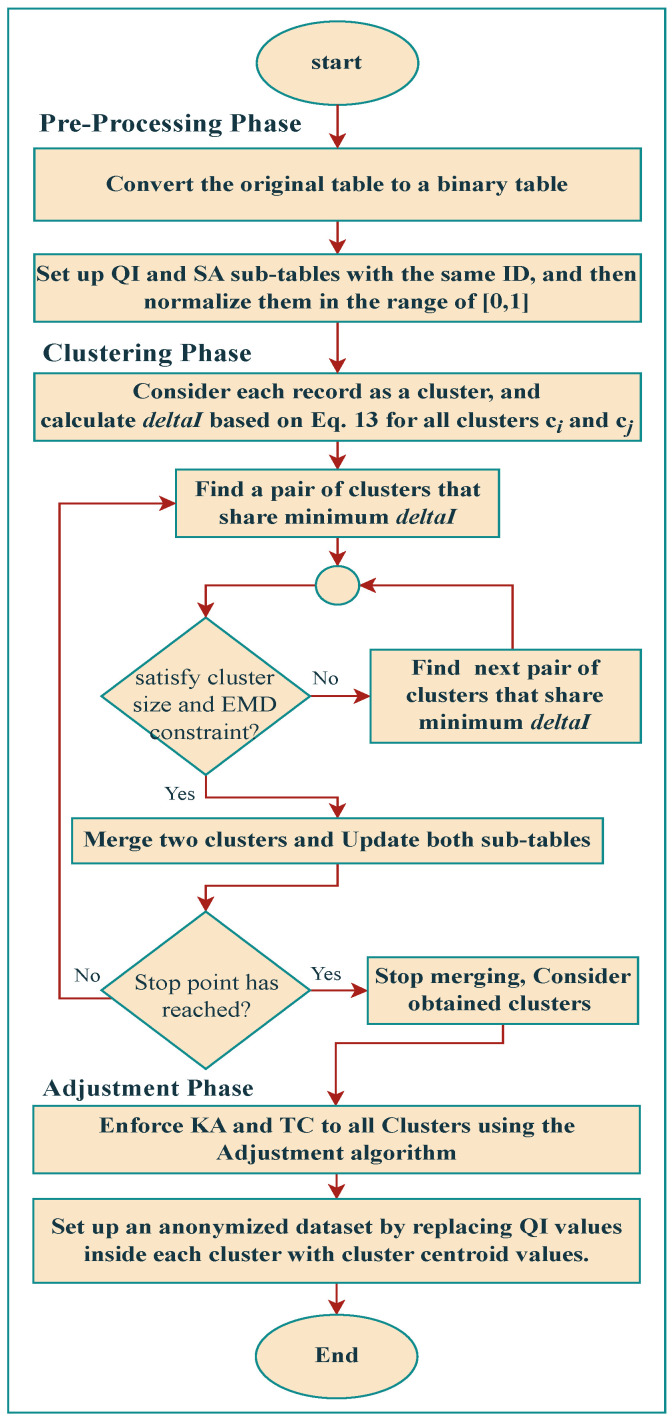
Overall flowchart of the proposed algorithm.

**Figure 5 entropy-25-01613-f005:**
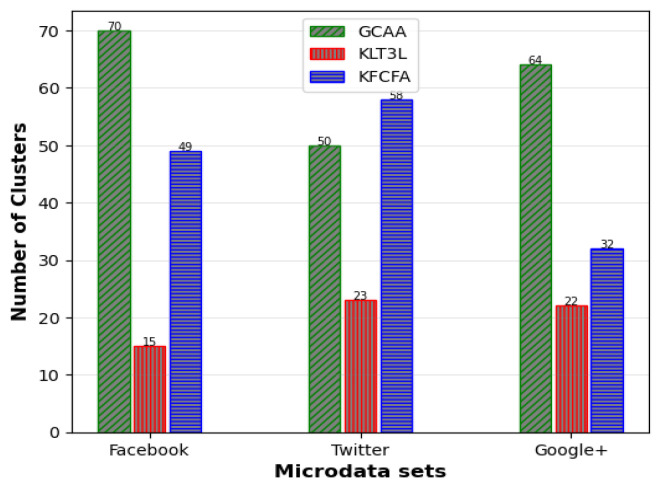
Comparison of C for simple privacy scenario.

**Figure 6 entropy-25-01613-f006:**
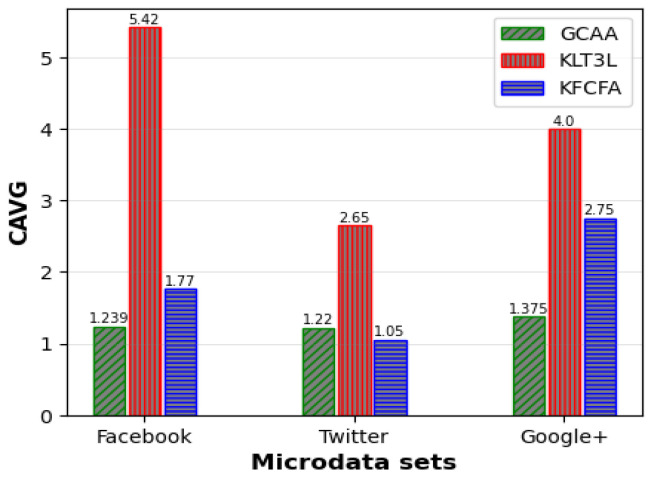
Comparison of CAVG for simple privacy scenario.

**Figure 7 entropy-25-01613-f007:**
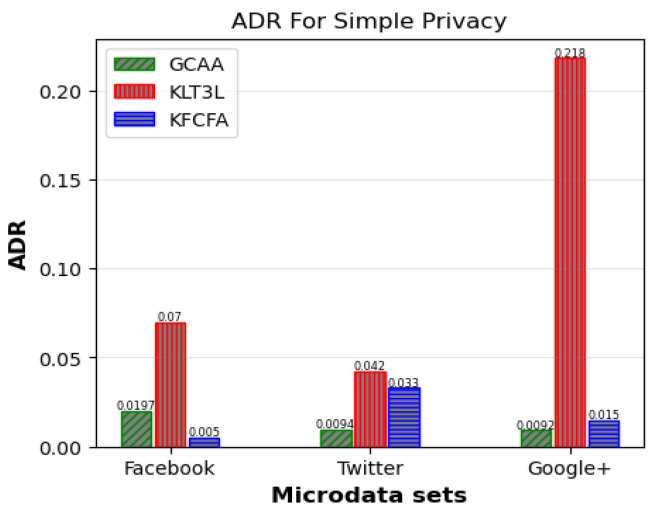
Comparison of ADR for simple privacy scenario.

**Figure 8 entropy-25-01613-f008:**
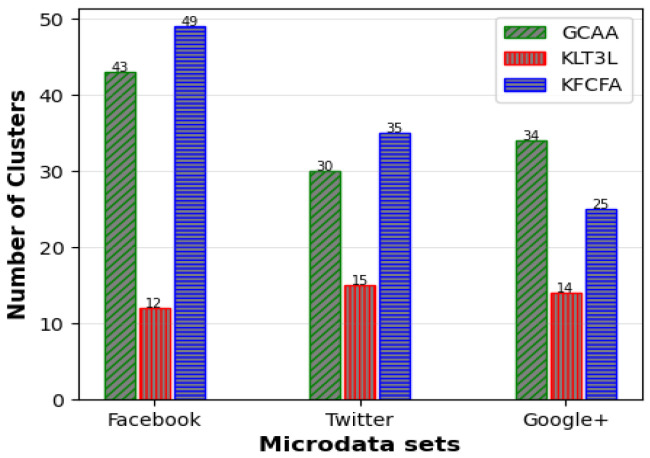
Comparison of C for medium privacy scenario.

**Figure 9 entropy-25-01613-f009:**
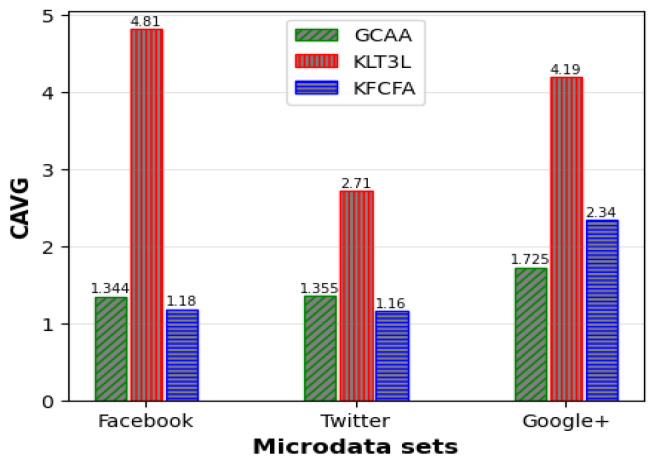
Comparison of CAVG for medium privacy scenario.

**Figure 10 entropy-25-01613-f010:**
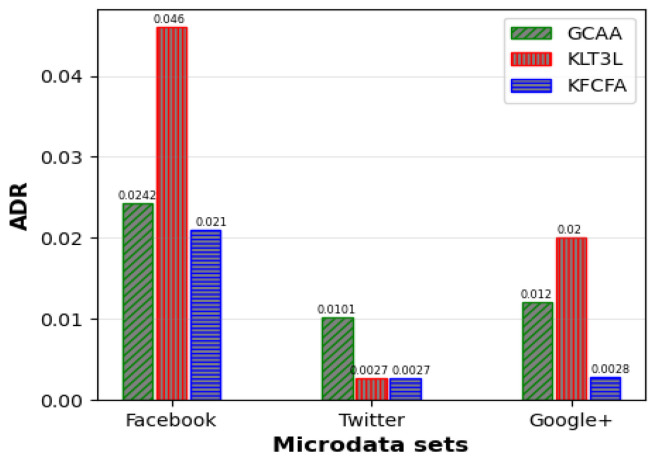
Comparison of ADR for medium privacy scenario.

**Figure 11 entropy-25-01613-f011:**
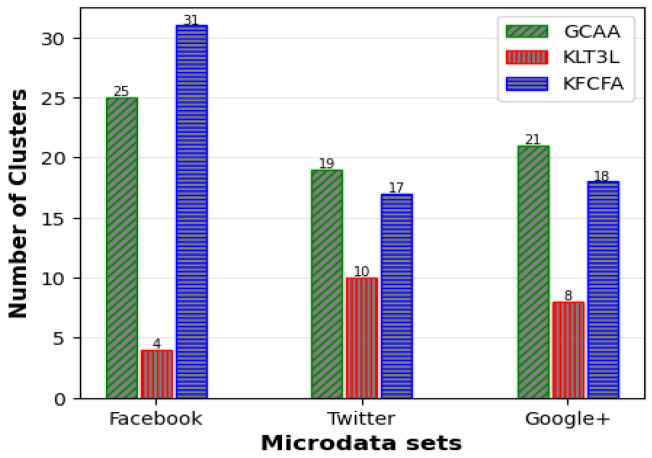
Comparison of C for strict privacy scenario.

**Figure 12 entropy-25-01613-f012:**
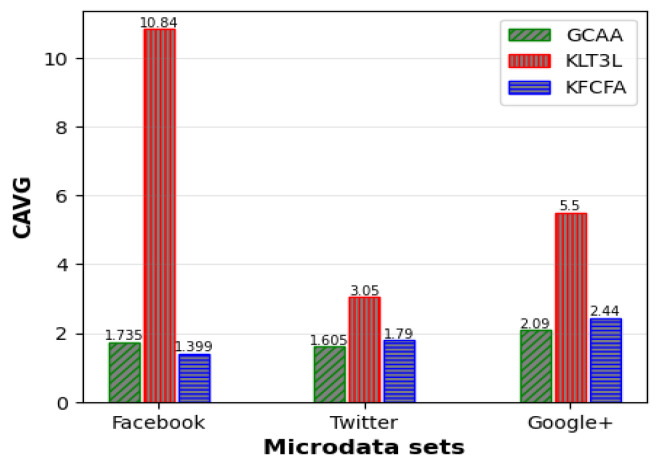
Comparison of CAVG for strict privacy scenario.

**Figure 13 entropy-25-01613-f013:**
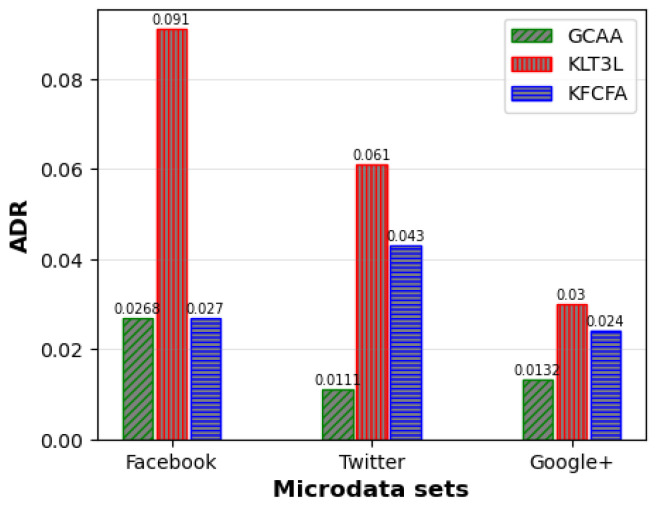
Comparison of ADR for strict privacy scenario.

**Figure 14 entropy-25-01613-f014:**
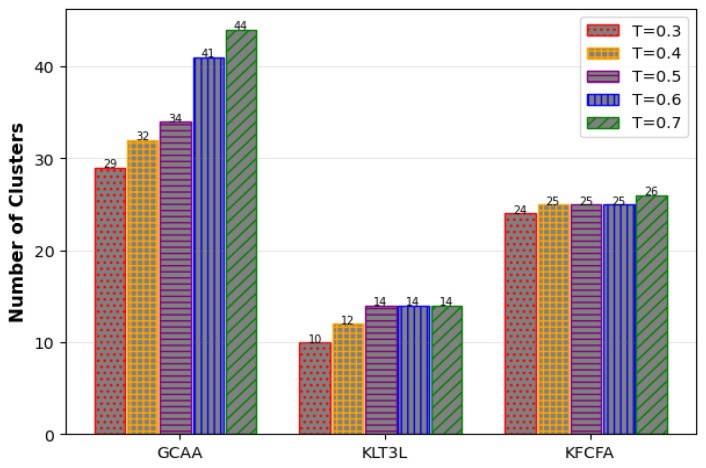
Comparison of the C for different values of *T* in the Google+ dataset.

**Figure 15 entropy-25-01613-f015:**
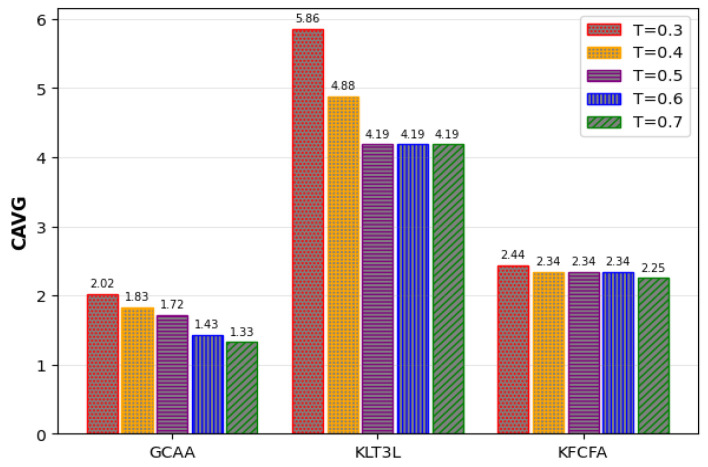
Comparison of the CAVG for different values of *T* in the Google+ dataset.

**Figure 16 entropy-25-01613-f016:**
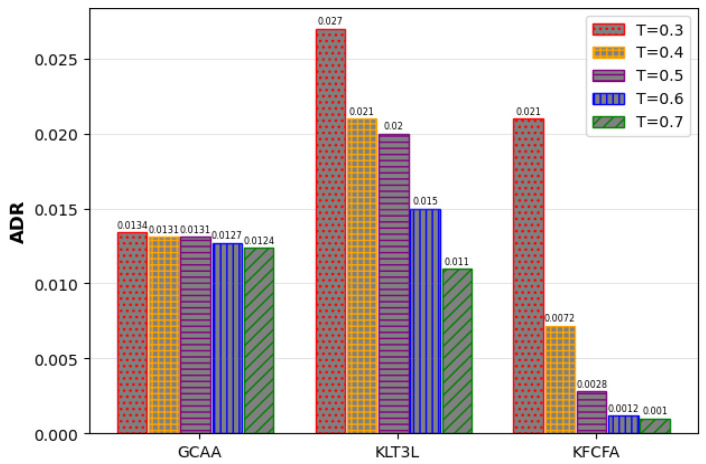
Comparison of the ADR for different values of *T* in the Google+ dataset.

**Table 1 entropy-25-01613-t001:** Patients’ original data.

No.	Weight	Sex	Age	Disease
1	73	Male	33	Pneumonia
2	56	Female	36	Pneumonia
3	82	Male	31	Pneumonia
4	71	Female	44	Bronchitis
5	51	Female	47	Colon cancer
6	68	Female	40	Flu
7	70	Female	55	Colitis
8	69	Male	60	Colon cancer
9	48	Female	59	Flu

**Table 2 entropy-25-01613-t002:** 3-Anonymity version of original data.

Cluster	Weight	Sex	Age	Disease
C1	70.33	Male	[30–39]	Pneumonia
	70.33	Male	[30–39]	Pneumonia
	70.33	Male	[30–39]	Pneumonia
C2	63.33	Female	[40–49]	Bronchitis
	63.33	Female	[40–49]	Colon cancer
	63.33	Female	[40–49]	Flu
C3	62.33	Female	[50–59]	Colitis
	62.33	Female	[50–59]	Colon cancer
	62.33	Female	[50–59]	Flu

**Table 3 entropy-25-01613-t003:** 3-Anonymity 2-Diversity version of original data.

Cluster	Weight	Sex	Age	Disease
C1	64.66	Female	[40–49]	Pneumonia
	64.66	Female	[40–49]	Colon cancer
	64.66	Female	[40–49]	Colitis
C2	57.33	Female	[40–49]	Pneumonia
	57.33	Female	[40–49]	Flu
	57.33	Female	[40–49]	Flu
C3	74.00	Male	[40–49]	Pneumonia
	74.00	Male	[40–49]	Colon cancer
	74.00	Male	[40–49]	Bronchitis

**Table 4 entropy-25-01613-t004:** A 0.33-Closeness version of original data.

Cluster	Weight	Sex	Age	Disease
C1	74.00	Male	[40–49]	Pneumonia
	74.00	Male	[40–49]	Colon cancer
	74.00	Male	[40–49]	Bronchitis
C2	61.00	Female	[40–49]	Pneumonia
	61.00	Female	[40–49]	Flu
	61.00	Female	[40–49]	Flu
	61.00	Female	[40–49]	Pneumonia
	61.00	Female	[40–49]	Colon cancer
	61.00	Female	[40–49]	Colitis

**Table 5 entropy-25-01613-t005:** The main abbreviation used in the paper.

Description	Abbreviation
K-Anonymity	KA
L-Diversity	LD
T-Closeness	TC
Information Loss	IL
Earth Mover’s Distance	EMD
Normalized Avarege QI-group	CAVG
Average Distortion Ratio	ADR
Greedy Clustering-based Anonymization Algorithm	GCAA

**Table 6 entropy-25-01613-t006:** The review of state-of-the-art privacy-preserving algorithms.

Paper	Privacy Model	Operation	Key Demerits
[29]	KA	Generalization	Insufficient to protect against attribute disclosure (AD) attacks High amount of IL
[33]	KA, LD	Microaggregation	Insufficient to protect against AD attacks
[22]	KA	Generalization	Insufficient to protect against AD attacks Overfitting to data
[23]	KA, LD, TC	Generalization	High amount of IL Low amount of data utility
[16]	KA	Generalization	Insufficient to protect against AD attacks
[24]	KA, LD, TC	Perturbation	High computational complexity Lack of scalability

**Table 7 entropy-25-01613-t007:** Original data table (*T*).

	Quasi-Identifiers			Sensitive Attributes	
**ID**	**Gender**	**Status**	**Nationality**	**Disease**	**Salary**
1	Male	Married	Russian	HIV	>50 K
2	Female	Single	American	Flu	>50 K
3	Female	Married	Iranian	Cancer	≤50 K
4	Male	Married	Iranian	HIV	≤50 K
5	Male	Single	American	Cancer	>50 K

**Table 8 entropy-25-01613-t008:** The normalized quasi-identifiers version ($T_{QI}$) of Table 7.

	The Quasi-Identifiers Values Domain						
**ID**	**Male**	**Female**	**Married**	**Single**	**Russian**	**American**	**Iranian**
1	1/3	0	1/3	0	1/3	0	0
2	0	1/3	0	1/3	0	1/3	0
3	0	1/3	1/3	0	0	0	1/3
4	1/3	0	1/3	0	0	0	1/3
5	1/3	0	0	1/3	0	1/3	0

**Table 9 entropy-25-01613-t009:** The normalized sensitive attribute version ($T_{SA}$) of Table 7.

	The Sensitive Attributes Values Domain				
**ID**	**HIV**	**Flu**	**Cancer**	**>50 K**	**≤50 K**
1	1/2	0	0	1/2	0
2	0	1/2	0	1/2	0
3	0	0	1/2	0	1/2
4	1/2	0	0	0	1/2
5	0	0	1/2	1/2	0

**Table 10 entropy-25-01613-t010:** Microdata sets characterstics.

Dataset	Number of Records	Number of Features	The Domain of Features Values
Facebook	347	19	224
Twitter	244	28	1364
Google+	352	6	289

**Table 11 entropy-25-01613-t011:** Different privacy scenarios.

Privacy Scenario	Parameter *k*	Parameter *t*
Simple Privacy	4	0.7
Medium Privacy	6	0.5
Strict Privacy	8	0.3

**Table 12 entropy-25-01613-t012:** The obtained result for a simple privacy scenario in the Facebook dataset.

Metrics	Description	GCAA (The Proposed Algorithm)	KLT3L	KFCFA	PSKA	FKMC
C	Number of Clusters	70	16	49	18	86
CAVG	Equation (Equation 13)	1.239	5.42	1.77	4.81	1.008
ADR	Equation (Equation 14)	0.0197	0.07	0.005	0.001	0.001
P${}_{KA}$	Penalty of KA	0	0	0	0	0
P${}_{TC}$	Penalty of TC	0	0	0	17	71
Time	Average Running Time	16	2.3	46	3.1	3.4

**Table 13 entropy-25-01613-t013:** The obtained result for a medium privacy scenario in the Facebook dataset.

Metrics	GCAA	KLT3L	KFCFA	PSKA	FKMC
C	43	12	49	17	57
CAVG	1.344	4.81	1.18	3.40	1.01
ADR	0.0242	0.046	0.021	0.001	0.001
P${}_{KA}$	0	0	0	0	0
P${}_{TC}$	0	0	0	17	37

**Table 14 entropy-25-01613-t014:** The obtained result for a strict privacy scenario in the Facebook dataset.

Metrics	GCAA	KLT3L	KFCFA	PSKA	FKMC
C	25	4	31	10	43
CAVG	1.735	10.84	1.399	4.33	1.0081
ADR	0.0268	0.091	0.027	0.001	0.001
P${}_{KA}$	0	0	0	0	0
P${}_{TC}$	0	0	0	10	43

## Data Availability

Data are contained within the article.

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
