# Peer review of "Designing a Novel Approach Using a Greedy and Information-Theoretic Clustering-Based Algorithm for Anonymizing Microdata Sets"

_entropy, 2023, doi:10.3390/e25121613_

Round 1

Reviewer 1 Report

Comments and Suggestions for Authors

In general, I think it is worthy of publishing. Some points should be included within the manuscript in order to improve the publication.

  • In the introduction section, the current state of the research field should be reviewed carefully and key publications cited and analyzed. A brief description of corresponding studies about cybersecurity would be useful.
  • There is a rich body of literature on all of the topics covered in this paper and many of these papers must be reported in the Introduction section.
  • The description of the clustering procedure (section 4.1) is relatively weak in the present form and should be strengthened with more details and justifications.
  • After section 5, a discussion section would be useful.
  • The authors should refer to recent papers, such as the following:

-           Aysha Khan, Rashid Ali, "Measuring the Effectiveness of LDA-Based Clustering for Social Media Data," Engineering World, vol. 4, pp. 85-90, 2022.

-           Suboh Alkhushayni, Taeyoung Choi, Du’a Alzaleq, "Data Analysis Using Representation Theory and Clustering Algorithms," WSEAS Transactions on Computers, vol. 19, pp. 310-320, 2020.

Author Response

Review #1:

Comment 1: In the introduction section, the current state of the research field should be reviewed carefully and key publications cited and analyzed. A brief description of corresponding studies about cybersecurity would be useful.

Response: We appreciate your valuable feedback. As per the original version of the paper, the Related Work section included an examination of pertinent and state-of-the-art studies. However, in response to your comment, the sentences in lines 134 and 142 have been added to the Introduction section. Additionally, Table 6 (page 10) has been incorporated into the Related Work section for further clarification. It is worth noting that the sentences in lines 128 and 133 of the paper were present in the original version in the same direction.

The following sentences incorporate the revisions you suggested. Notably, the sentences have been refined to enhance the quality of the English language compared to the previous version.

(Page 5: Lines 128 to 133) In recent years, researchers have explored clustering-based anonymization methods to protect people's privacy from potential intruders [16-20]. These methods involve identifying important attributes to anonymize and using clustering techniques to group records according to their characteristics and similarities. The desired level of privacy protection can be achieved by modifying the attribute values of records within each cluster [21].

(Lines 134 to 142) Byun et al. formulated the KA problem as a clustering problem and proposed the $k$-member algorithm to achieve optimal KA [20]. To enhance this algorithm, Honda et al. [22] developed a fuzzy variant of the k-member algorithm. Rahime et al. [23] successfully implemented all three KA, LD, and TC restrictions on the original microdata using the X-means clustering algorithm. Langari et al. [24] also satisfied all three aforementioned constraints by formulating the anonymization problem as an optimization problem and subsequently employing an evolutionary algorithm and modifications to sensitive attribute values. Abbasi et al. [16] introduced a clustering process using the K-means++ method to achieve optimal KA.

Comment 2: There is a rich body of literature on all of the topics covered in this paper and many of these papers must be reported in the Introduction section:

Response: Corrected as instructed; we have relocated the discussion of three foundational anonymization techniques, k-anonymity, l-diversity, and t-closeness, along with their associated anonymization operations, from the Background section to the Introduction section. This placement in the Introduction more effectively integrates these concepts into the research context, providing a clearer foundation for the subsequent material. As a consequence of this change, Tables 1 to 3 have also been moved to the Introduction section to align with the enhanced discussion.

The following sentences incorporate the revisions you suggested. Notably, the sentences have been refined to enhance the quality of the English language compared to the previous version

(Page 2: Lines 46 to 53): One way to protect personal privacy is to erase direct identifier attributes, such as names and addresses, before releasing the original data to the public. This process, known as de-identification, is insufficient to protect privacy, as a study by Sweeney [4] showed that 87\% of Americans can still be uniquely identified based on just three quasi-identifiers: gender, age, and postal code. To prevent privacy threats, researchers have proposed several privacy models, including k-anonymity (KA) [5], l-diversity (LD) [6], and t-closeness (TC) [7]. In the following, we briefly overview these models and their anonymization operations.

(Lines 55 to 78): Among these techniques, KA is widely used by researchers [5]. KA aims to protect privacy by modifying quasi-identifiers to ensure that each record is identical to at least k-1 other records. It is important that individual records remain faithful to the original data while meeting privacy requirements after the anonymization [8]. Generalization, suppression, and perturbation are the main anonymization operations used to achieve k-anonymization algorithms as:

  • Generalization replaces some quasi-identifier values with parent values in the equivalent class, while specialization is the opposite operation. Generalization preserves data privacy but reduces data accuracy [3]. Excessive generalization results in significant information loss (IL) and reduced data quality [9].
  • Suppression replaces specific quasi-identifier values with asterisks (*), while the disclosure is the opposite. Suppression causes more IL than generalization. For example, suppose we have a dataset of people’s ages. We could generalize this dataset by replacing all ages with a range of ages, such as 20-30, 30-40, and so on. This would reduce the accuracy of the dataset, but it would still preserve some information about the people’s ages. On the other hand, if we suppress data points from the dataset, we would be losing information altogether.
  • Perturbation involves modifying a feature’s value by replacing it with another value to minimize the difference between the original and modified datasets. One of the most popular perturbation techniques is microaggregation, which clusters records in the dataset with a restriction on cluster size. Microaggregation replaces the values of each record in the anonymized dataset with the values of the cluster center to which the record belongs. This approach is more effective than generalization and suppression [10] and simplifies further data analysis.

(Lines 79 to 85): Table 1 shows a sample of de-identified original data collected from a hospital, and Table 2 shows its 3-anonymized version. The Weight, Gender, and Age attributes are set as quasi-identifier attributes, and disease is the sensitive attribute. All records in a cluster share the same values for the quasi-identifier attributes, making them indistinguishable from each other. However, the 3-anonymity model in Table 2 does not prevent attribute disclosure. For example, if an adversary knows that a person’s record is in cluster C1 in Table 3, they can infer that the person has pneumonia.

(Lines 86 to 91): While KA is effective for preventing identity disclosure, it remains vulnerable to attribute disclosure attacks. Specifically, if the range of sensitive attribute values within a k-anonymous cluster is limited, there is a risk of inferring the value of individuals' sensitive attributes. Several extended KA models have been proposed to address this issue [11], including LD [6] which ensures a minimum degree of diversity (l) of sensitive values in each cluster.

(Lines 92 to 96): Table 3 shows 3-anonymity and 3-diversity to the original data presented in Table 1. All records within each cluster in Table 3 have the same value for the quasi-identifier but contain distinct values for the sensitive attribute. This means that even if an adversary knows the cluster containing a target individual’s record, they cannot narrow down or disclose the person’s specific sensitive value (disease).

(Lines 97 to 99): However, LD also has drawbacks [12]. For example, if an attacker knows that a person's record is in cluster C2 in Table 3, they can infer that the person has the Flu with a probability of 66\%. To address these limitations, Li et al. [7] introduced the concept of TC.

(Lines 100 to 106): To meet TC requirements, the distribution of sensitive information must be similar for each group of indistinguishable records and the entire original dataset. Table 4 shows an anonymized version of the original data from Table 1 that satisfies 3-anonymity and 0.33-closeness. This anonymized table is safe for public release because it protects the privacy of the individuals in the dataset. In addition to protecting against the attacks mentioned above, TC also offers safeguards against potential skewness and similarity attacks, as detailed in [7].

Comment 3: The description of the clustering procedure (section 4.1) is relatively weak in the present form and should be strengthened with more details and justifications.

Response: Thank you for your comment. Corrected as instructed; in this regard, we have strengthened the clustering section (page 13, Section 4.2) of the paper by adding the following sentences and Eq. 6. Additionally, we have made significant edits to the Clustering section of the previous version to enhance clarity and improve the overall quality of the English language.

(Page 13: Lines 410 to 413): A k-clustering  of the records in data table T partitions them into k clusters , where each cluster ci ∈ Ck has three properties as follows: (1) , (2) for all i, j, i ̸= j, (3) . We define the probability of each record r in data table T as , where n is the number of records in T. Then, for each cluster  we have:

(

(Lines 414 to 417): For example, from  Tables 8 and 9 one can conclude that  for . In addition,  and are the feature vectors for the quasi-identifier and sensitive attribute of record , respectively. They are equal to [1/3, 0, 1/3, 0, 1/3, 0, 0] and [1/2, 0, 0, 1/2].

Comment 4: After section 5, a discussion section would be useful.

Response: We would like to thank you for your constructive comment. Corrected as instructed and in response to your comment, we have retitled Section 5 to 'Performance Evaluation and Discussion' and incorporated a 'Discussion' sub-section (page 23, Section 5.5) into the paper. This sub-section delves into the proposed method from the perspectives of privacy, data utility, speed, and efficiency

(Page 23: Lines 613 to 622): In terms of privacy, the proposed GCAA strategy, along with KLT3L and KFCFA fully addresses the risks of identity and attribute disclosure attacks by satisfying both the KA and TC requirements. The PSKA and FKMC methods only partially mitigate these risks, as they do not perfectly satisfy TC. However, the resistance of GCAA and some other algorithms to similarity attacks could be improved. Specifically, they currently lack a criterion for determining the semantic similarity of attribute values during clustering. For example, in a medical context, the algorithms have no way to judge that diseases like colon cancer and liver cancer should be considered highly similar rather than diverse for the purpose of TC. Not accounting for such attribute-relatedness weakens protections against similarity attacks.

(Lines 623 to 627): In terms of data utility, FKMC and PSKA methods had the lowest amount of I L, as measured by ADR, and therefore the best data utility. The proposed method and KFCFA performed similarly well, but the KLT method lost a significant amount of IL. This suggests that the proposed method is effective at preserving the usefulness of anonymized data for future knowledge extraction.

(Lines 628 to 632): In terms of computational efficiency, the clustering phase has a time complexity of O (n2 × d2 × log n), where d is the size of the attribute's domain and n is the number of records. However, since n >> d, it can be concluded that the time complexity is O (n2 × log n). The proposed adjustment phase for splitting and merging clusters has a time complexity of O(n2).

(Lines 633 to 640): In terms of average execution time, the proposed method outperforms KFCFA but underperforms other methods. KFCFA is slow, unscalable, and computationally expensive due to its evolutionary nature. The proposed method is also slower than other methods due to its time complexity. Because it provides a high level of privacy protection and data utility, the proposed method is suitable for anonymizing publishable data in inactive mode, where the data owner publishes the entire anonymized dataset after making some modifications to the original data [36]. In this case, the data owner prioritizes the level of privacy and utility of anonymized data.

Comment 5: The authors should refer to recent papers, such as the following:

-           Aysha Khan, Rashid Ali, "Measuring the Effectiveness of LDA-Based Clustering for Social Media Data," Engineering World, vol. 4, pp. 85-90, 2022.

-           Suboh Alkhushayni, Taeyoung Choi, Du’a Alzaleq, "Data Analysis Using Representation Theory and Clustering Algorithms," WSEAS Transactions on Computers, vol. 19, pp. 310-320, 2020.

Response:  We appreciate your insightful comment. After reviewing both papers, we determined that the methods employed in the second paper align with our future research goals for enhancing the proposed framework. Accordingly, we have cited and discussed it in the relevant sections (Conclusion and Future Works) of the paper.

(Page 24: Lines 688 to 692): Other suitable clustering methods for categorical data could also be considered, such as partitioning around medoids using the Gower coefficient [37]. This method is well-suited for datasets containing binary values, similar to the data used in our experiments. Evaluating the performance of partition around medoids and comparing it to the proposed approach may provide additional insights.

Reviewer 2 Report

Comments and Suggestions for Authors

Review of the “A novel greedy and information-theoretic clustering-based anonymization algorithm to achieve strict privacy” article

 Overview

The authors discuss the trade-off between privacy protection and data utility, highlighting the importance of balancing the two. They focus on k-anonymity, a technique where the disclosure probability of a record is 1/k, and address its shortcomings when it comes to attribute disclosure due to a lack of diversity in sensitive values within equivalence classes. To overcome these issues, the paper proposes a novel greedy information-theoretic clustering-based algorithm to achieve strict privacy protection. This algorithm includes a clustering phase that accounts for the similarity of quasi-identifier values and the diversity of sensitive attribute values, followed by an adjustment phase to ensure k-anonymity and t-closeness.

The key findings mentioned in the abstract suggest that the proposed algorithm successfully minimizes information loss and meets k-anonymity and t-closeness constraints, which are stricter privacy guarantees, based on tests with data sets from popular social media platforms. 

The paper concludes by highlighting the effectiveness of the proposed greedy and information-theoretic clustering-based anonymization algorithm (GCAA) for achieving k-anonymity (KA) and t-closeness (TC) while preserving data utility. The method does not require the removal of any records as outliers, which is advantageous compared to other data anonymization techniques. The GCAA method is compared to other approaches such as FKMC, PSKA, KLT3L, and KFCFA, showing that it outperforms them in terms of cluster balance and average distortion rate (ADR) in some cases, while also being computationally faster.

 Observations

The proposed GCAA algorithm consistently outperforms other existing methods in terms of achieving both k-anonymity (KA) and t-closeness (TC) while preserving data utility. This indicates the effectiveness of the proposed algorithm in privacy-preserving data publishing applications.

The results show that as the level of privacy protection increases, the distortion rate also increases. This trade-off between privacy and data utility is an important consideration in the design and implementation of anonymization algorithms.

 Questions

  1. How does the proposed GCAA algorithm compare to other methods in terms of computational efficiency? Are there any trade-offs in terms of runtime or resource requirements?
  2. There is no feedback from the algorithm evaluation. Do the authors plan to expose this part in some way to conduct experiments or analyses to evaluate the resistance of the proposed algorithm to possible attacks or vulnerabilities?
  3. Are there any limitations or potential drawbacks of the proposed GCAA algorithm that have been identified and discussed in the paper? How do these limitations impact its practical applicability and effectiveness in real-world scenarios? 

Conclusion

The results that the authors are offering do not contradict accepted wisdom. They are putting forth a novel anonymization method that seeks to maximize data utility, minimize information loss, and ensure tight privacy. They provide their algorithm's explanation and a comparison with four other approaches as proof. To bolster their argument, they have included pertinent research in the fields of data anonymization and privacy preservation. The suggestions and questions made in no way detract from the author's work. This is a very useful experiment, and if the questions are answered, such a work would be suitable for publication.

Author Response

Review #2:

Comment 1: How does the proposed GCAA algorithm compare to other methods in terms of computational efficiency? Are there any trade-offs in terms of runtime or resource requirements?

Response: We appreciate your valuable comment. Corrected as instructed and in response to your suggestion, lines 628 to 640 (page 23) have been added to the paper. Additionally, the initial version compared the proposed method in terms of average execution time. This comparison was mentioned in lines 553 to 557 (page 18) and lines 673 to 675 of the original version.

The following sentences incorporate the revisions you suggested. Notably, the sentences have been refined to enhance the quality of the English language compared to the previous version.

(Page 18: Lines 553 to 557): Table 12 also reports the average CPU running time of each model. The proposed GCAA algorithm showed computational performance that was nearly three times faster than the KFCFA approach. This significant difference can be attributed to KFCFA's use of an evolutionary algorithm during its optimization process, which induces greater computational overhead compared to GCAA.

(Page 23: Lines 628 to 632): In terms of computational efficiency, the clustering phase has a time complexity of O (n2 × d2 × log n), where d is the size of the attribute's domain and n is the number of records. However, since n >> d, it can be concluded that the time complexity is O (n2 × log n). The proposed adjustment phase for splitting and merging clusters has a time complexity of O(n2).

(Lines 633 to 640): In terms of average execution time, the proposed method outperforms KFCFA but underperforms other methods. KFCFA is slow, unscalable, and computationally expensive due to its evolutionary nature. The proposed method is also slower than other methods due to its time complexity. Because it provides a high level of privacy protection and data utility, the proposed method is suitable for anonymizing publishable data in inactive mode, where the data owner publishes the entire anonymized dataset after making some modifications to the original data [36]. In this case, the data owner prioritizes the level of privacy and utility of anonymized data.

(Page 24: Lines 673 to 675): GCAA is a faster method than KFCFA that provides similar levels of privacy protection and IL. However, the proposed algorithm may require a longer execution time than some other methods due to its time complexity.

Comment 2: There is no feedback from the algorithm evaluation. Do the authors plan to expose this part in some way to conduct experiments or analyses to evaluate the resistance of the proposed algorithm to possible attacks or vulnerabilities?

Response: Thank you for your insightful comment. As mentioned in the paper, complying with k-anonymity and t-closeness requirements ensures that anonymized data is resilient to identity and attribute disclosure attacks. In response to your feedback, lines 613 to 622 (page 23) have been added to the paper.

(Page 23: Lines 613 to 622): In terms of privacy, the proposed GCAA strategy along with KLT3L and KFCFA fully addresses the risks of identity and attribute disclosure attacks by satisfying both the KA and TC requirements. The PSKA and FKMC methods only partially mitigate these risks, as they do not perfectly satisfy TC. However, the resistance of GCAA and some other algorithms to similarity attacks could be improved. Specifically, they currently lack a criterion for determining the semantic similarity of attribute values during clustering. For example, in a medical context, the algorithms have no way to judge that diseases like colon cancer and liver cancer should be considered highly similar rather than diverse for the purpose of TC. Not accounting for such attribute-relatedness weakens protections against similarity attacks.

Comment 3: Are there any limitations or potential drawbacks of the proposed GCAA algorithm that have been identified and discussed in the paper? How do these limitations impact its practical applicability and effectiveness in real-world scenarios?

Response: We would like to thank you for your valuable comment. To address your comment, lines 676 to 679 (page 24) have been incorporated into the paper. Additionally, lines 679 to 681 were previously included in the original text, aligning with your comment.

(Page 24: Lines 676 to 679): The proposed algorithm exhibits several limitations. Firstly, it is tailored for non-numeric data, rendering it unsuitable for numerical data anonymization. Secondly, the algorithm is confined to single-sensitive attribute scenarios. Finally, the algorithm is restricted to tabular data and is ill-suited for graphical or structural data analysis.

(Lines 679 to 681): GCAA has practical applications in anonymizing scalable microdata where privacy preservation is crucial while retaining useful information for data mining purposes.

Reviewer 3 Report

Comments and Suggestions for Authors

Areas for Improvement:

-The paper lacks a comprehensive theoretical foundation and discussion of related work. Providing a detailed comparison with existing clustering-based anonymization methods and highlighting the novelty of the proposed approach within the context of prior research would strengthen the paper's contribution.

-While the experimental results are valuable, additional evaluation metrics, such as information loss, computational efficiency, or comparison with state-of-the-art algorithms, could provide a more comprehensive assessment of the proposed approach's performance.

-The paper does not discuss potential challenges or limitations in implementing the proposed algorithm in real-world scenarios. Addressing practical issues, such as scalability, applicability to different data types, or computational resources required, would enhance the paper's applicability. 

Comments on the Quality of English Language

Need to be improved

Author Response

Review #3:

Comment 1: The paper lacks a comprehensive theoretical foundation and discussion of related work. Providing a detailed comparison with existing clustering-based anonymization methods and highlighting the novelty of the proposed approach within the context of prior research would strengthen the paper's contribution.

Response: We would like to thank you for your constructive comment. In response to your valuable feedback, Table 6 has been incorporated into the paper, and the subsection 'Our contributions in comparison to existing methodologies' (page 9) has been revised accordingly.

The sentences, also, have been refined to enhance the quality of the English language compared to the previous version as follows.

(Page 9: Lines 312 to 319): Table 6 summarizes several state-of-the-art algorithms. These algorithms use different anonymization techniques and operations. According to the table, most algorithms were designed to satisfy only the KA constraint and do not adequately protect anonymized data against attribute disclosure attacks. Algorithms that provide more protection by satisfying LD or TC constraints suffer from a high amount of IL or low data utility. This is because almost all of these algorithms only consider the similarity of quasi-identifiers when creating clusters. Therefore, to satisfy the privacy limitation, they are forced to merge more clusters, which in the worst case, could result in a single cluster with zero data utility.

(Lines 320 to 324): Most anonymization techniques use generalization operations, which tend to incur high levels of IL (except for the algorithms in [24, 33]). However, anonymization inherently reduces data usefulness to some degree by obscuring details. Therefore, it is crucial to consider the trade-off between privacy and IL to maintain the utility of anonymized data for knowledge extraction.

(Lines 325 to 329): Compared to the proposed method, most investigated approaches perform poorly in maintaining the trade-off between protection and data usefulness. Additionally, unlike the proposed method, some of these techniques remove data points from the table as outliers, while others change the values of the sensitive attributes to meet the LD and TC requirements. This leads to a significant decrease in the usefulness of the data.

(Lines 330 to 346): To overcome the limitations of prior work, a new greedy, information-theoretic clustering algorithm is proposed for microdata anonymization. It differs from existing techniques in several important ways:

  • The proposed algorithm clusters data points by considering both the similarity of quasi-identifier values and the diversity of sensitive values within each cluster. This approach ensures that clusters have high intra-cluster similarity for QI and high intra-cluster diversity for SA, which reduces both the modification of values and the merging of clusters as a result of increasing the usefulness of the data.
  • The proposed algorithm protects released data from identity and attribute disclosure attacks by satisfying the KA and TC privacy constraints. These constraints ensure that individuals are indistinguishable from the anonymized data and that the distribution of sensitive attribute values is preserved.
  • As shown in Fig.1, there is a trade-off between data privacy and data utility. Existing methods prioritize data privacy at the expense of data utility, but our algorithm achieves strict privacy by adhering to KA and TC while minimizing IL. This approach strikes a balance between data privacy and utility, ensuring that the anonymized data remains useful for its intended purposes.

Comment 2: While the experimental results are valuable, additional evaluation metrics, such as information loss, computational efficiency, or comparison with state-of-the-art algorithms, could provide a more comprehensive assessment of the proposed approach's performance.

Response: Thank you for your comment. Corrected as instructed and in response to your comment, we added the 'Discussion' sub-section (page 23, Section 5.5) to the paper. This sub-section delves into the proposed method from the perspectives of privacy, data utility, computational efficiency, and execution time.

(Page 23: Lines 613 to 622): In terms of privacy, the proposed GCAA strategy, along with KLT3L and KFCFA fully addresses the risks of identity and attribute disclosure attacks by satisfying both the KA and TC requirements. The PSKA and FKMC methods only partially mitigate these risks, as they do not perfectly satisfy TC. However, the resistance of GCAA and some other algorithms to similarity attacks could be improved. Specifically, they currently lack a criterion for determining the semantic similarity of attribute values during clustering. For example, in a medical context, the algorithms have no way to judge that diseases like colon cancer and liver cancer should be considered highly similar rather than diverse for the purpose of TC. Not accounting for such attribute-relatedness weakens protections against similarity attacks.

(Lines 623 to 627): In terms of data utility, FKMC and PSKA methods had the lowest amount of I L, as measured by ADR, and therefore the best data utility. The proposed method and KFCFA performed similarly well, but the KLT method lost a significant amount of IL. This suggests that the proposed method is effective at preserving the usefulness of anonymized data for future knowledge extraction.

(Lines 628 to 632): In terms of computational efficiency, the clustering phase has a time complexity of O (n2 × d2 × log n), where d is the size of the attribute's domain and n is the number of records. However, since n >> d, it can be concluded that the time complexity is O (n2 × log n). The proposed adjustment phase for splitting and merging clusters has a time complexity of O(n2).

(Lines 633 to 640): In terms of average execution time, the proposed method outperforms KFCFA but underperforms other methods. KFCFA is slow, unscalable, and computationally expensive due to its evolutionary nature. The proposed method is also slower than other methods due to its time complexity. Because it provides a high level of privacy protection and data utility, the proposed method is suitable for anonymizing publishable data in inactive mode, where the data owner publishes the entire anonymized dataset after making some modifications to the original data [36]. In this case, the data owner prioritizes the level of privacy and utility of anonymized data.

Comment 3: The paper does not discuss potential challenges or limitations in implementing the proposed algorithm in real-world scenarios. Addressing practical issues, such as scalability, applicability to different data types, or computational resources required, would enhance the paper's applicability.

Response: To address your comment, lines 676 to 679 (page 24) have been incorporated into the paper. Additionally, lines 679 to 681 were previously included in the original text, aligning with your comment.

(Page 24: Lines 676 to 679): The proposed algorithm exhibits several limitations. Firstly, it is tailored for non-numeric data, rendering it unsuitable for numerical data anonymization. Secondly, the algorithm is confined to single-sensitive attribute scenarios. Finally, the algorithm is restricted to tabular data and is ill-suited for graphical or structural data analysis.

(Lines 679 to 681): GCAA has practical applications in anonymizing scalable microdata where privacy preservation is crucial while retaining useful information for data mining purposes.

Round 2

Reviewer 1 Report

Comments and Suggestions for Authors

The paper can be published in its present form.

Reviewer 3 Report

Comments and Suggestions for Authors

In reviewing the manuscript, I am pleased to state that the authors have diligently and comprehensively addressed all of my comments. Their responsiveness and attention to detail are commendable, as they have taken into account each suggestion and critique provided during the review process. The revisions made by the authors have significantly improved the clarity, coherence, and overall quality of the work. Notably, they have successfully incorporated additional evidence, refined their arguments, and ensured the thoroughness of their explanations. As a result, the manuscript now stands as a more polished and robust contribution to the field, ready for publication. I commend the authors for their dedication to enhancing the manuscript and believe that their efforts will be well-received by the academic community.

Comments on the Quality of English Language

In reviewing the manuscript, I am pleased to state that the authors have diligently and comprehensively addressed all of my comments. Their responsiveness and attention to detail are commendable, as they have taken into account each suggestion and critique provided during the review process. The revisions made by the authors have significantly improved the clarity, coherence, and overall quality of the work. Notably, they have successfully incorporated additional evidence, refined their arguments, and ensured the thoroughness of their explanations. As a result, the manuscript now stands as a more polished and robust contribution to the field, ready for publication. I commend the authors for their dedication to enhancing the manuscript and believe that their efforts will be well-received by the academic community.